# Mitochondria from osteolineage cells regulate myeloid cell-mediated bone resorption

Peng Ding [1,2,4], Chuan Gao[1,2,4], Jian Zhou[1,2,4], Jialun Mei[1,2,4], Gan Li[1,2], Delin Liu [1,2], Hao Li[1,2], Peng Liao [1,2], Meng Yao[1,2], Bingqi Wang[1,2], Yafei Lu[1,2], Xiaoyuan Peng [1], Chenyi Jiang[1], Jimin Yin[1], Yigang Huang[1], Minghao Zheng [3], Youshui Gao [1] ✉, Changqing Zhang [1,2] ✉ & Junjie Gao [1,2] ✉

Interactions between osteolineage cells and myeloid cells play important roles in maintaining skeletal homeostasis. Herein, we find that osteolineage cells transfer mitochondria to myeloid cells. Impairment of the transfer of mitochondria by deleting MIRO1 in osteolineage cells leads to increased myeloid cell commitment toward osteoclastic lineage cells and promotes bone resorption. In detail, impaired mitochondrial transfer from osteolineage cells alters glutathione metabolism and protects osteoclastic lineage cells from ferroptosis, thus promoting osteoclast activities. Furthermore, mitochondrial transfer from osteolineage cells to myeloid cells is involved in the regulation of glucocorticoid-induced osteoporosis, and glutathione depletion alleviates the progression of glucocorticoid-induced osteoporosis. These findings reveal an unappreciated mechanism underlying the interaction between osteolineage cells and myeloid cells to regulate skeletal metabolic homeostasis and provide insights into glucocorticoid-induced osteoporosis progression.

The skeletal system contains a series of bone and blood lineage cells, and their commitments and dynamic interactions play indispensable roles in the maintenance of skeletal homeostasis[1–3]. Osteolineage cells arising from mesenchymal stem cells (MSCs) and myeloid cells arising from hematopoietic stem cells (HSCs) are the two primary functional cell lineages[2,4]. Osteolineage cells, including osteoprogenitors, osteoblastic cells, and osteocytes, are mainly responsible for bone formation, and have been increasingly shown to be the important regulators of bone and bone marrow, which have been reported to regulate the bone marrow hematopoietic niche and influence myeloid and lymphoid lineages and their progeny commitments[5–8]. In addition, myeloid cells can differentiate into osteoclastic lineage cells for bone resorption, and osteoclastic lineage cells spatiotemporally interact with osteolineage cells during bone remodeling[9,10]. Disorders of

specific energy metabolism of these cell lineages lead to an imbalance in skeletal homeostasis, resulting in the occurrence and development of bone-related diseases[11]. While the complex communications between these cell lineages have been documented, the regulatory mechanisms are not fully understood.

Mitochondria play important roles in the regulation of cell lineage preservation and specification[2,12–14], and have emerged as key contributors to cell homeostasis, particularly in the skeletal microenvironment[15–17]. Different cell types gain functionally and molecularly distinct mitochondrial types as they mature during development[18–20], and different cell types transferring mitochondria to regulate neighboring or remote cells have been observed[21–28]. Cell-to-cell transferred mitochondria, as dynamic, energy-transforming, and signaling organelles, have been increasingly recognized as important

[1]Department of Orthopaedics, Shanghai Sixth People's Hospital Affiliated to Shanghai Jiao Tong University School of Medicine, 200233 Shanghai, China. [2]Institute of Microsurgery on Extremities, Shanghai Sixth People's Hospital Affiliated to Shanghai Jiao Tong University School of Medicine, 200233 Shanghai, China. [3]Centre for Orthopaedic Translational Research, Medical School, University of Western Australia, Nedlands, WA 6009, Australia. [4]These authors contributed equally: Peng Ding, Chuan Gao, Jian Zhou, Jialun Mei. ✉e-mail: gaoyoushui@sjtu.edu.cn; zhangcq@sjtu.edu.cn; colingjj@163.com

information process systems[29] that regulate the cascade amplification mechanism of many molecular signals within the recipient cell, thus controlling gene transcription and translation and ultimately determining the phenotype of the cell[29,30]. However, whether this process occurs in the skeletal system or regulates skeletal metabolic homeostasis in vivo has not been fully elucidated. We hypothesize that osteolineage cells regulate other cell lineages in the skeletal system via intercellular mitochondrial transfer.

Herein, we found that mitochondria from osteolineage cells regulate myeloid cell-mediated bone resorption by altering glutathione (GSH) metabolism-regulated ferroptosis. Impairment of mitochondrial transfer from osteolineage cells by deleting MIRO1 promoted myeloid cell commitment to osteoclastogenesis and subsequent bone loss. Furthermore, mitochondrial transfer from osteolineage cells to myeloid cells also contributed to glucocorticoid-induced osteoporosis (GIOP), and GSH depletion alleviated the progression of GIOP. Collectively, our study suggests that intercellular mitochondrial transfer between osteolineage cells and myeloid cells represents an unappreciated mechanism that regulates skeletal metabolic homeostasis and provides insights into GIOP progression.

## Results

### Osteolineage cells transfer mitochondria to myeloid cells

To depict the transportation of osteolineage mitochondria to bone marrow cells, we employed osteolineage-specific mitochondria reporter mice with Dendra2 expressed under the control of promoters active in specific osteolineage cell mitochondria, in which *Prrx1^cre MitoDendra^+/+* marks osteoprogenitor mitochondria[31], *Col1a1^cre/ERT2 MitoDendra^+/+* marks osteoblastic cell mitochondria[32] and *Dmp1^cre MitoDendra^+/+* marks osteocyte mitochondria[33]. We first analyzed *Prrx1*, *Col1a1* and *Dmp1* expression in each cluster of bone marrow cells based on our single-cell RNA sequencing data[4] and demonstrated that there was no obvious expression of *Prrx1*, *Col1a1*, and *Dmp1* in other clusters of bone marrow cells except the MSC cluster (Supplementary Fig. 1a), which confirmed the specificity of our mitochondria reporter mice. We then collected bone marrow cells to perform flow cytometry (Fig. 1a). After removing adherent and dead cells, we further clustered bone marrow cells into three main lineage cells: hematopoietic stem and progenitor cells (HSPCs), myeloid lineage cells and lymphoid lineage cells, which give rise to several subclusters including monocytes/macrophages, neutrophils, dendritic cells (DCs), CD4+ T cells, CD8+ T cells and B cells (Supplementary Fig. 1b), and cells containing Dendra2 signals (Dendra2 positive) were recognized as mitochondria recipient cells.

We first analyzed mitochondrial transfer in the *Prrx1^cre MitoDendra^+/+* reporter mice and there were mitochondria transferred to CD11b+ myeloid cells (Fig. 1b, accounting for approximately 1.21% of myeloid cells), B220+ lymphoid cells (Fig. 1c, accounting for approximately 2.23% of lymphoid cells) and HSPCs (Fig. 1d, accounting for approximately 1.72% of HSPCs), in which osteoprogenitors transferred the most mitochondria to myeloid cells (accounting for 60.47%), followed by lymphoid cells (accounting for 34.88%) (Fig. 1e). HSPCs received the lowest number of mitochondria (accounting for 4.65%) (Fig. 1e). Further subcluster analysis demonstrated that osteoprogenitors transferred mitochondria to monocytes/macrophages (Fig. 1f, g), neutrophils (Fig. 1i) and DCs (Supplementary Fig. 1c) instead of CD4+ T cells (Supplementary Fig. 1d) and CD8+ T cells (Supplementary Fig. 1e). Considering that osteoprogenitors transferred the most mitochondria to myeloid cells, we performed further analysis and found that monocytes/macrophages received the most transferred mitochondria among myeloid cells (accounting for 50.89%) (Fig. 1h).

We then examined the mitochondrial transfer network in the *Col1a1^cre/ERT2 MitoDendra^+/+* reporter mice and the *Dmp1^cre MitoDendra^+/+* reporter mice. Mitochondria were observed to transfer to CD11b+

myeloid cells, B220+ lymphoid cells and HSPCs in the *Col1a1^cre/ERT2 MitoDendra^+/+* reporter mice (Supplementary Fig. 2a–c). Subcluster analysis demonstrated that monocytes/macrophages (Supplementary Fig. 2d, e), neutrophils (Supplementary Fig. 2f) and DCs (Supplementary Fig. 2g) instead of CD4+ (Supplementary Fig. 2h) and CD8+ T cells (Supplementary Fig. 2i) received the transferred mitochondria. Intriguingly, myeloid cells, especially monocytes/macrophages, received the most transferred mitochondria (Supplementary Fig. 2j, k) which was consistent with the findings in the *Prrx1^cre MitoDendra^+/+* reporter mice. Similarly, mitochondrial transfer was also observed in CD11b+ myeloid cells, B220+ lymphoid cells and HSPCs of the *Dmp1^cre MitoDendra^+/+* reporter mice (Supplementary Fig. 2l–n). Monocytes/macrophages (Supplementary Fig. 2o, p), neutrophils (Supplementary Fig. 2q) and DCs (Supplementary Fig. 2r) instead of CD4+ (Supplementary Fig. 2s) and CD8+ T cells (Supplementary Fig. 2t) were the target recipient cells of transferred mitochondria. Moreover, myeloid cells, especially monocytes/macrophages received the most transferred mitochondria (Supplementary Fig. 2u, v).

However, the number of monocytes/macrophages, B cells and neutrophils in the *Dmp1^cre MitoDendra^+/+* reporter mice that received transferred mitochondria was significantly lower than that in the *Col1a1^cre/ERT2 MitoDendra^+/+* reporter mice and the *Prrx1^cre MitoDendra^+/+* reporter mice (Fig. 1j). Collectively, these data depicted the mitochondrial transfer network from osteolineage cells to other lineage cells residing in the bone marrow, of which monocytes/macrophages receive the most mitochondria from osteolineage cells.

### Osteolineage cells transfer mitochondria to osteoclastic cells

Considering that osteolineage *MitoDendra^+/+* reporter mice exhibited the most mitochondrial transfer in myeloid lineage cells, especially monocytes/macrophages clusters, and monocytes/macrophages could differentiate into osteoclastic lineage cells to participate in bone remodeling[2], we then examined whether osteolineage cell-derived mitochondria regulate myeloid cell commitment to osteoclastic lineage cells. As osteoblasts are the main functional cells of osteolineage cells that build new bone and couple bone remodeling with osteoclastic lineage cells[2,34], we extracted both long bone and calvaria osteoblasts (OBs) from *Prrx1^cre MitoDendra^+/+* mice and bone marrow derived macrophages (BMMs) from *mT/mG* mice as previously reported[4,35,36]. BMMs were induced toward osteoclast precursors (OCPs) with macrophage colony-stimulating factor (M-CSF, 100 ng/mL) and receptor activator of nuclear factor kappa-B ligand (RANKL, 100 ng/mL) stimulation. Meanwhile, OBs were added and coincubated for two days, and then, flow cytometry was performed. Flow cytometry analysis showed that both long bone and calvaria OBs could transfer mitochondria to osteoclastic lineage cells and approximately 10% mitochondria were transferred to osteoclastic lineage cells (Fig. 2a and Supplementary Fig. 3a, b). Osteoblastic cell line MC3T3-E1 stained with MitoTracker could also transfer mitochondria to osteoclastic lineage cells (Supplementary Fig. 3c), indicating the conservative phenomenon of transferring mitochondria from osteolineage cells to osteoclastic lineage cells. As long bone and calvaria OBs share similar transcriptomes[37], we used calvaria OBs in following in vitro experiments. We also extracted and cocultured *Lyz2^cre MitoDendra^+/+* OCPs and *mT/mG* OBs. However, no obvious mitochondrial transfer was observed from *Lyz2^cre MitoDendra^+/+* OCPs to *mT/mG* OBs (Fig. 2b), implying that mitochondrial transfer between osteolineage cells and osteoclastic lineage cells is unidirectional. Moreover, with the prolongation of the coculture time, *mT/mG* OCPs revealed incremental mitochondrial transfer from *Prrx1^cre MitoDendra^+/+* OBs (Supplementary Fig. 3d, e), indicating that mitochondrial transfer from OBs to OCPs is time dependent. Together, these data confirmed the unidirectional mitochondrial transfer from osteolineage cells to osteoclastic lineage cells. To investigate whether this type of unidirectional

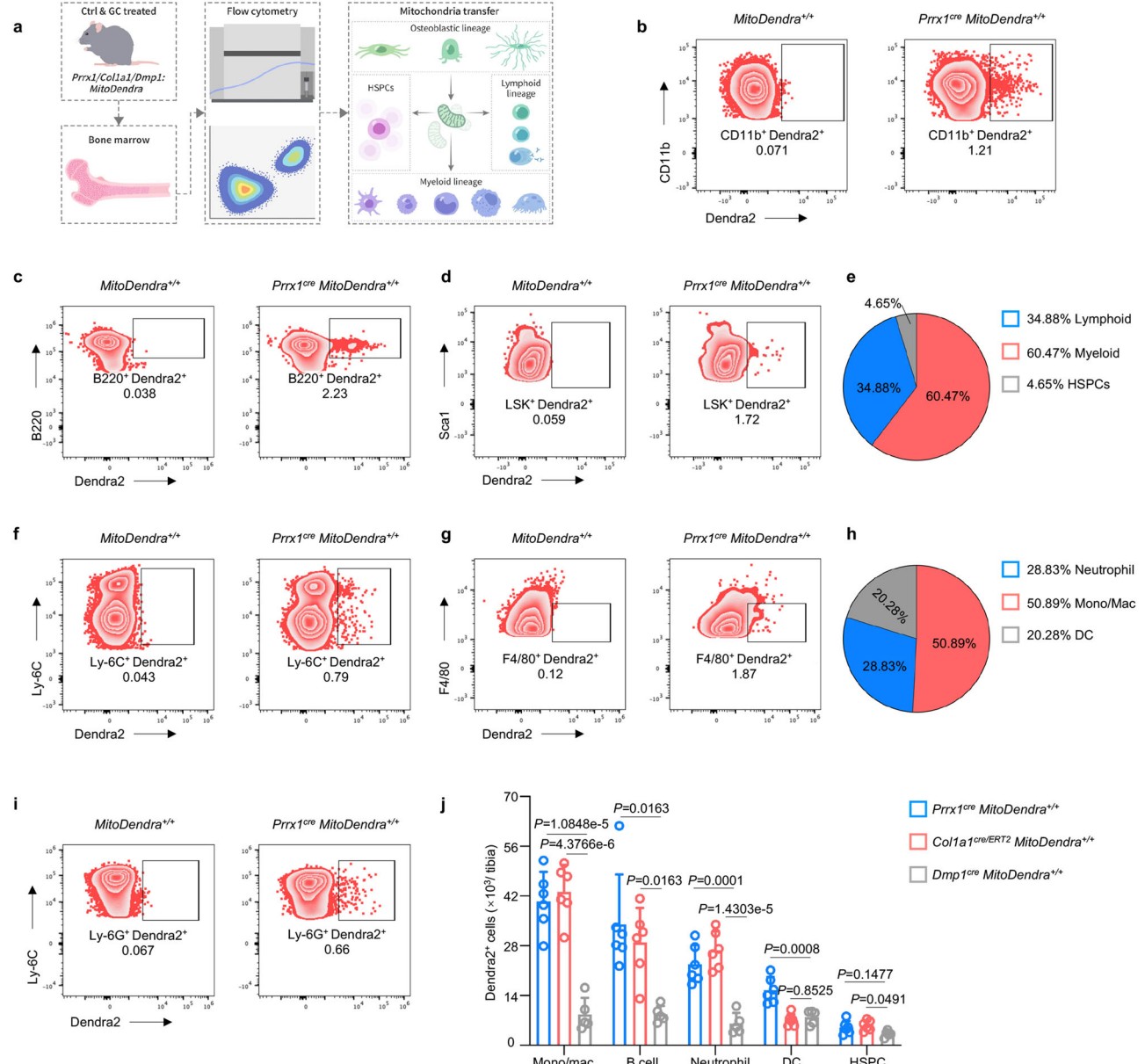

**Fig. 1 | Osteolineage cells transfer mitochondria to myeloid cells. a** Schematic showing the experimental design. Bone marrow cells of osteolineage-specific mitochondria reporter mice (*Prrx1^cre MitoDendra^+/+*, *Col1a1^cre/ERT2 MitoDendra^+/+* and *Dmp1^cre MitoDendra^+/+* mice) were collected and stained, and then, flow cytometry was performed to evaluate the mitochondrial transfer to other lineage cells. This figure was created by P.D. and cartoonized by Mr. Zihao Li. **b–d** Representative images of flow cytometry showing mitochondrial transfer to CD11b^+ myeloid cells (**b**), B220^+ lymphoid cells (**c**) and HSPCs (**d**) in the *Prrx1^cre MitoDendra^+/+* mice (*n* = 6 per group). **e** The proportion of cell types that received the osteolineage cell-derived mitochondria in the *Prrx1^cre MitoDendra^+/+* mice, of which myeloid cells received the most transferred mitochondria (*n* = 6 per group). **f, g** Representative images of flow cytometry showing mitochondrial transfer to monocytes (**f**) and macrophages (**g**) in the *Prrx1^cre MitoDendra^+/+* mice (*n* = 6 per group). **h** The proportion of subclusters of myeloid cells that received osteolineage cell-derived mitochondria in the *Prrx1^cre MitoDendra^+/+* mice, of which monocytes/macrophages received the most transferred mitochondria (*n* = 6 per group). **i** Representative images of flow cytometry showing mitochondrial transfer to neutrophils (*n* = 6 per group). **j** Flow cytometry analysis showing the number of mitochondrial transferred from osteolineage cells to other lineage cells, including monocytes/macrophages, B cells, neutrophils, DCs and HSPCs in vivo (*n* = 6 in the *Prrx1^cre MitoDendra^+/+* and the *Col1a1^cre/ERT2 MitoDendra^+/+* group and *n* = 5 in the *Dmp1^cre MitoDendra^+/+* group), demonstrating decreased mitochondrial transfer to monocytes/macrophages, B cells and neutrophils in the *Dmp1^cre MitoDendra^+/+* mice compared to the *Prrx1^cre MitoDendra^+/+* and *Col1a1^cre/ERT2 MitoDendra^+/+* mice. Data are presented as the mean ± s.d., with biologically individual data points shown. *P* values were determined by nonparametric one-way ANOVA test with Dunn's multiple comparisons (B cell of **j**), and ordinary one-way ANOVA test with Tukey's multiple comparisons of others (**j**). Source data are provided as a Source Data file.

mitochondrial transfer occurred in myeloid cells committed to osteoclast differentiation, we then cocultured OBs and OCPs with RANKL stimulation from Day 1 to Day 5. With RANKL stimulation, increased mitochondrial transfer from OBs to OCPs was observed, in which the number of OCPs that received mitochondria increased almost 5 times from Day 1 to Day 5 (Fig. 2c, d), indicating the essential

role of mitochondria from osteolineage cells during myeloid cells toward osteoclast differentiation.

Previous studies have reported the mitochondrial transfer from donor to recipient cells via extracellular vesicles[23,38]. To assess this possibility, we utilized the Transwell system with *Prrx1^cre MitoDendra^+/+* OBs plated in the upper well and *mT/mG* OCPs plated in the lower well,

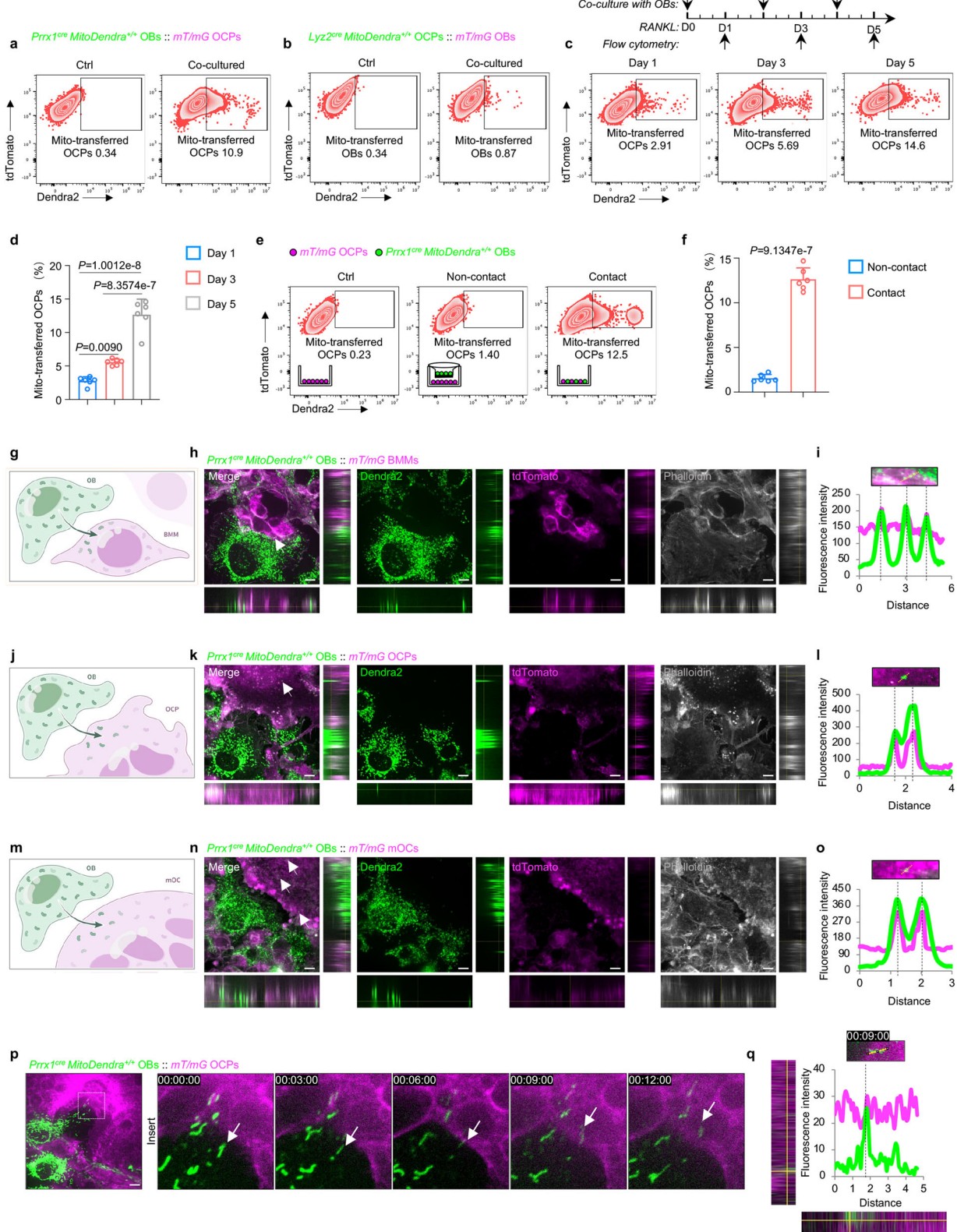

as illustrated in Fig. 2e. However, little mitochondrial transfer was observed in noncontact group compared to the contact group (Fig. 2e, f), indicating the importance of direct contact for transferring mitochondria from osteolineage cells to osteoclastic lineage cells. To determine whether mitochondria are internalized into the cell cytoplasm, we then further performed super-resolution fluorescence microscopy on cocultures between *Prrx1$^{cre}$ MitoDendra$^{+/+}$* OBs and *mT/*

*mG* osteoclastic lineage cells at different stages from BMMs, and OCPs to mature osteoclasts (mOCs) (Fig. 2g, j, m), to observe the mito-chondria distribution and dynamic morphological alterations. Inter-estingly, we found that OBs underwent morphological changes when interacting with osteoclastic lineage cells, as confirmed by a more enriched mitochondria distribution on the side close to the osteo-clastic lineage cells than on the side away from the osteoclastic lineage

**Fig. 2 | Osteolineage cells transfer mitochondria to osteoclastic cells.**
**a** Representative images of flow cytometry demonstrating the transfer of Dendra2-positive mitochondria from *Prrx1^cre MitoDendra^+/+* calvarial OBs to *mT/mG* osteoclast precursors (OCPs). **b** Representative images of flow cytometry demonstrating almost no transfer of Dendra2-positive mitochondria from *Lyz2^cre MitoDendra^+/+* OCPs to *mT/mG* OBs. **c, d** Representative images of flow cytometry (**c**) and analysis (**d**) showing the increased transfer of Dendra2-positive mitochondria from OBs to OCPs with RANKL stimulation from Day 1, Day 3 to Day 5 (*n* = 6 per group). **e, f** Representative images of flow cytometry (**e**) and analysis (**f**) showing the transfer of Dendra2-positive mitochondria via direct contact (*n* = 6 per group). This figure was created and cartoonized by J.J.G. **g** Schematic of OBs transferring mitochondria to BMMs. This figure was created by P.D. and cartoonized by Mr. Zihao Li. **h, i** Representative images (**h**) and fluorescence intensity analysis (**i**) showing the colocalization of Dendra2-positive mitochondria and BMMs

cytoplasm. Scale bar, 25 μm. **j** Schematic of OBs transferring mitochondria to OCPs. This figure was created by P.D. and cartoonized by Mr. Zihao Li. **k, l** Representative images (**k**) and fluorescence intensity analysis (**l**) showing the colocalization of Dendra2-positive mitochondria and OCPs cytoplasm. Scale bar, 25 μm. **m** Schematic of OBs transferring mitochondria to mOCs. This figure was created by P.D. and cartoonized by Mr. Zihao Li. **n, o** Representative images (**n**) and fluorescence intensity analysis (**o**) showing the colocalization of Dendra2-positive mitochondria and mOCs cytoplasm. Scale bar, 25 μm. **p, q** Time lapse videography (**p**) and fluorescence intensity analysis (**q**) showing the colocalization of Dendra2-positive mitochondria during movement towards adjacent OCPs. Scale bar, 25 μm. Data are presented as the mean ± s.d., with biologically individual data points shown. *P* values were determined by ordinary one-way ANOVA with Tukey's multiple comparisons test (**d**) and unpaired two-tailed Student's *t* test with Welch's correction (**f**). Source data are provided as a Source Data file.

cells (Fig. 2h, k, n), which indicated that OBs tend to transfer mitochondria to surrounding cells. Furthermore, OBs-derived mitochondria were detected in the cytoplasm of BMMs (Fig. 2h, i), OCPs (Fig. 2k, l) and mOCs (Fig. 2n, o). In addition, time lapse videography and fluorescence intensity analysis revealed a mitochondria uptake event, trafficking of OB-derived mitochondria and internalization into OCPs (Fig. 2p, q). However, the mitochondria distribution of OCPs was not changed when interacting with OBs, and no mitochondria were transferred to OBs (Supplementary Fig. 3f, g). Together, these data demonstrated that osteolineage cells transfer mitochondria to osteoclastic lineage cells.

## Osteolineage mitochondria inhibit osteoclast activity

To examine the role of osteolineage cell-derived mitochondria during myeloid cells commitment toward osteoclastic lineage cells, we first separated OCPs with or without direct and indirect coculture to investigate osteoclast activity change. Intriguingly, we found that when OCPs were directly cocultured with OBs under RANKL stimulation, osteoclast activity was significantly inhibited (Supplementary Fig. 3h). When OCPs were indirectly cocultured with OBs under RANKL stimulation, osteoclast activity was significantly promoted (Supplementary Fig. 3h). As osteolineage cells transfer mitochondria to osteoclastic lineage cells via direct contact, we investigated whether osteolineage cell-derived mitochondria exert inhibitory effects on osteoclastogenesis. We first sorted myeloid cells which have received or not osteoprogenitor mitochondria and performed unbiased RNA sequencing to investigate the impact of mitochondrial transfer from osteolineage cells to myeloid cells in vivo. As revealed by RNA sequencing, Gene ontology (GO) analysis demonstrated the main enrichment of immune-related processes (Supplementary Fig. 3i) and top 10 Kyoto Encyclopedia of Genes and Genomes (KEGG) analyses showed that the hematopoietic cell lineage pathway was enriched (Supplementary Fig. 3j), indicating that mitochondrial transfer from osteolineage cells could alter myeloid cell commitments, thus regulating immune responses.

To further uncover the impact of altered myeloid cell commitments on osteoclastogenesis by mitochondrial transfer, we then cocultured *Prrx1^cre MitoDendra^+/+* OBs and *mT/mG* OCPs, and we sorted *mT/mG* mtD2^pos (OCPs that had internalized mitochondria from OBs) and mtD2^neg OCPs (OCPs that had not taken up mitochondria from OBs), combined with OCPs without coculture (Ctrl), and performed unbiased RNA sequencing (Fig. 3a). Principal component analysis (PCA) indicated that OCPs that had internalized mitochondria from OBs were transcriptionally distinct from OCPs that had not taken up mitochondria (Fig. 3b). Analysis of the top 100 differentially expressed genes revealed that the transcriptional pattern of the mtD2^pos group was different from that of the Ctrl and mtD2^neg groups (Fig. 3c). Furthermore, a volcano plot of the differentially expressed genes between the mtD2^pos and mtD2^neg groups showed that 416 genes were significantly upregulated and 2450 genes were significantly

downregulated (Fig. 3d). The top 20 differentially expressed genes in mtD2^pos versus mtD2^neg in OCPs were mainly downregulated osteoclastic genes, including *Ctsk, Oscar, Dcstamp, Atp6v0d2, Mst1r, Car2* and *Ccr1* (Fig. 3d). Moreover, the top 10 GO analyses revealed that differentially expressed genes caused by mitochondrial transfer were enriched in positive regulation of bone resorption, osteoclast differentiation, positive regulation of osteoclast differentiation and multinuclear osteoclast differentiation (Fig. 3e), which was consistent with the top 10 KEGG analyses showing that the osteoclast differentiation pathway was enriched (Fig. 3f). Moreover, gene set enrichment analysis (GSEA) indicated that OCPs that had received mitochondria from OBs were enriched in genes associated with osteoclast differentiation (Fig. 3g), cell activities (Supplementary Fig. 3k) and metabolism (Supplementary Fig. 3l). In addition, we assessed *Csf1r* and *Tnfrsf11a* expression (which are the most important surface receptors on osteoclastic lineage cells), and there was no significant difference between mtD2^+ and mtD2^- cells (Supplementary Fig. 3m). These data indicate that osteolineage cell-derived mitochondria exert inhibitory effects on osteoclastic lineage cells independent of ligand-receptor interactions.

Subsequent qPCR verified that genes critical for osteoclastogenesis, including *Acp5, Ctsk, Atp6v0d2, Dcstamp, Oscar,* and *Nfatc1,* were significantly downregulated in the OCPs that had received mitochondria compared to those in the mtD2^neg group (Fig. 3h). Furthermore, sorted mtD2^pos and mtD2^neg OCPs were induced toward osteoclastogenesis, and a bone resorption assay was performed. The number of mature osteoclasts (Fig. 3I, j) and bone resorption area (Fig. 3k, l) in the mtD2^pos group were significantly lower than those in the mtD2^neg group. Collectively, these data demonstrated that osteolineage cell-derived mitochondria inhibit myeloid cell commitments to osteoclast differentiation and subsequent bone resorption.

## MIRO1 mediates the intercellular mitochondrial transfer

Previous studies have reported that MIRO1, as a calcium-binding GTPase, plays a critical role in regulating mitochondrial transport[39–42]. We then examined whether MIRO1 participates in regulating mitochondrial transfer from osteolineage cells to osteoclastic lineage cells. We first examined MIRO1 expression in OBs cocultured with OCPs with or without RANKL stimulation. There was no significant difference in MIRO1 expression in OBs cocultured with OCPs after RANKL stimulation (Supplementary Fig. 4a). We then downregulated MIRO1 expression in OBs (Fig. 4a), and impaired mitochondrial transfer from OBs to OCPs was observed (Fig. 4b, c). Pharmacological inhibitors, namely, 6-thio-GTP (inhibits Rac1 GTPase) and ML141 (inhibits Cdc42), also inhibited the mitochondrial transfer process (Fig. 4d, e) at noncytotoxic concentrations, as previously reported[43]. Furthermore, we upregulated MIRO1 expression in OBs (Fig. 4f), and mitochondrial transfer from OBs to OCPs was strengthened (Fig. 4g, h). Together, these data demonstrated that MIRO1 mediates the mitochondrial transfer process from osteolineage cells to osteoclastic lineage cells.

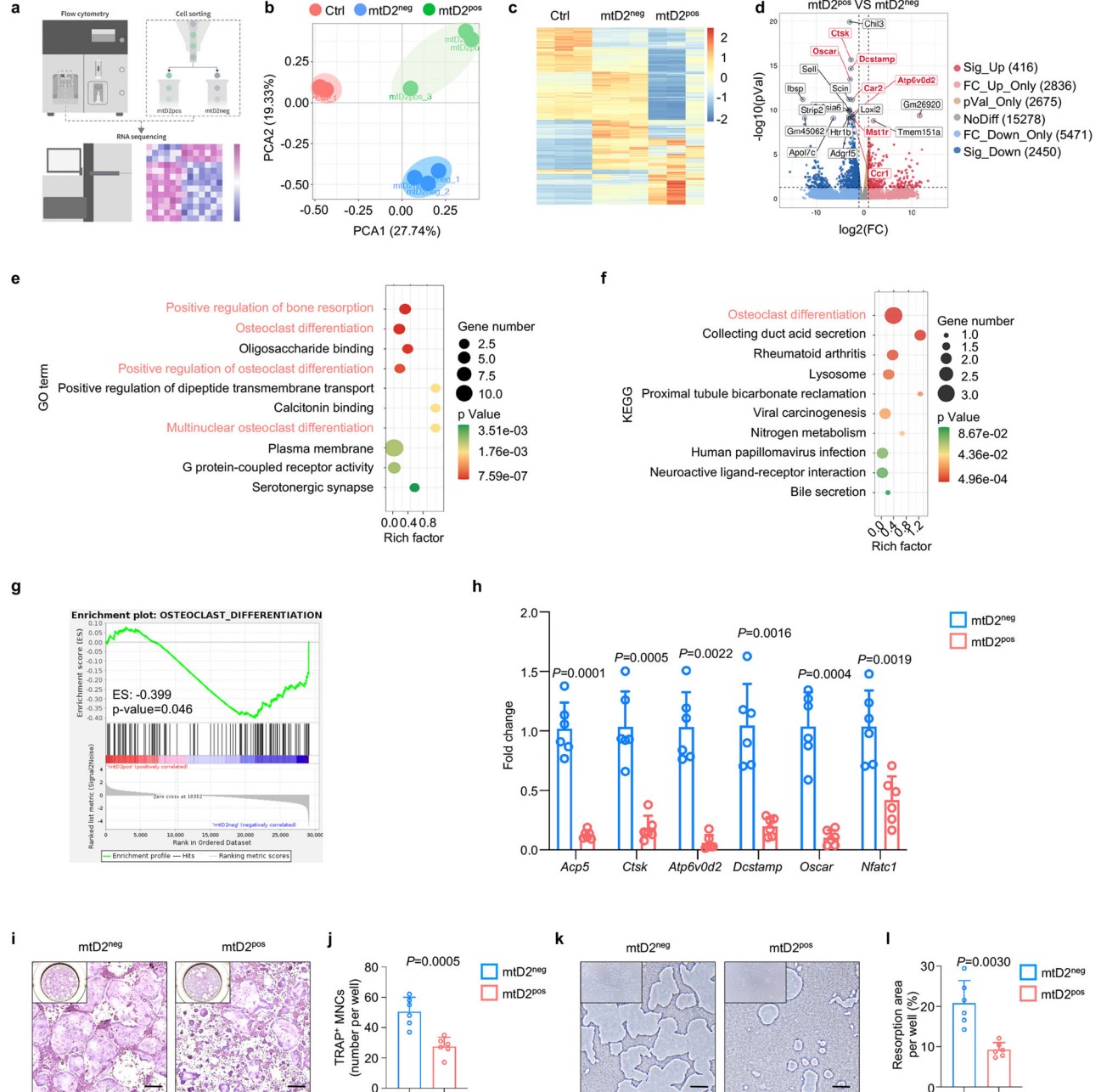

**Fig. 3 | Osteolineage cell-derived mitochondria inhibit osteoclast activities.**
**a** Schematic showing the experimental design. tdTomato+Dendra2+ cells were sorted as mtD2pos cells. tdTomato+Dendra2- cells were sorted as mtD2neg cells. OCPs without coculture were named Ctrl. This figure was created by P.D. and cartoonized by Mr. Zihao Li. **b** PCA of RNA sequencing of mtD2neg, mtD2pos and Ctrl OCPs, showing obvious separation between different groups. **c** Heatmap analysis of RNA sequencing of mtD2neg, mtD2pos and Ctrl OCPs. **d** Volcano plot of the top 20 significantly differentially expressed genes between mtD2neg and mtD2pos OCPs, showing that osteoclast-related genes were downregulated after receiving osteolineage cells derived mitochondria. **e** The top 10 GO terms of differentially expressed genes caused by mitochondrial transfer, demonstrating enrichment mainly in positive regulation of bone resorption, osteoclast differentiation, positive regulation of osteoclast differentiation and multinuclear osteoclast differentiation. **f** The top 10 KEGG pathways showing that osteoclast differentiation pathway was enriched. **g** GSEA showing the enrichment of genes associated with osteoclast

differentiation. **h** qPCR verification of the expression of osteoclast signature genes at the mRNA level between mtD2neg and mtD2pos OCPs (*n* = 6 per group). **i, j** TRAP staining of in vitro osteoclastogenesis from sorted mtD2neg and mtD2pos OCPs (**i**) and quantitative analysis (**j**) of TRAP positive cells (nucleus > 3) per well, demonstrating impaired osteoclastogenesis after receiving osteolineage cell-derived mitochondria (*n* = 6 per group). Scale bar, 250 μm. TRAP denotes tartrate-resistant acid phosphatase. **k, l** Bone resorption assay of sorted mtD2neg and mtD2pos OCPs (**k**) and quantification of resorption area per well (**l**), demonstrating impaired bone resorption after receiving osteolineage cell-derived mitochondria (*n* = 6 per group). Scale bar, 250 μm. Data are presented as the mean ± s.d., with biologically individual data points shown. *P* values were determined by two-tailed wald test (**d**), hypergeometric test (**e, f**), two-tailed permutation test (**g**), two-tailed Mann–Whitney *U* test (*Atp6v0d2* of **h**), unpaired two-tailed Student's *t* test with Welch's correction (*Acp5, Ctsk, Dcstamp, Oscar* of **h, l**), and unpaired two-tailed Student's *t* test of others (*Nfatc1* of **h, j**). Source data are provided as a Source Data file.

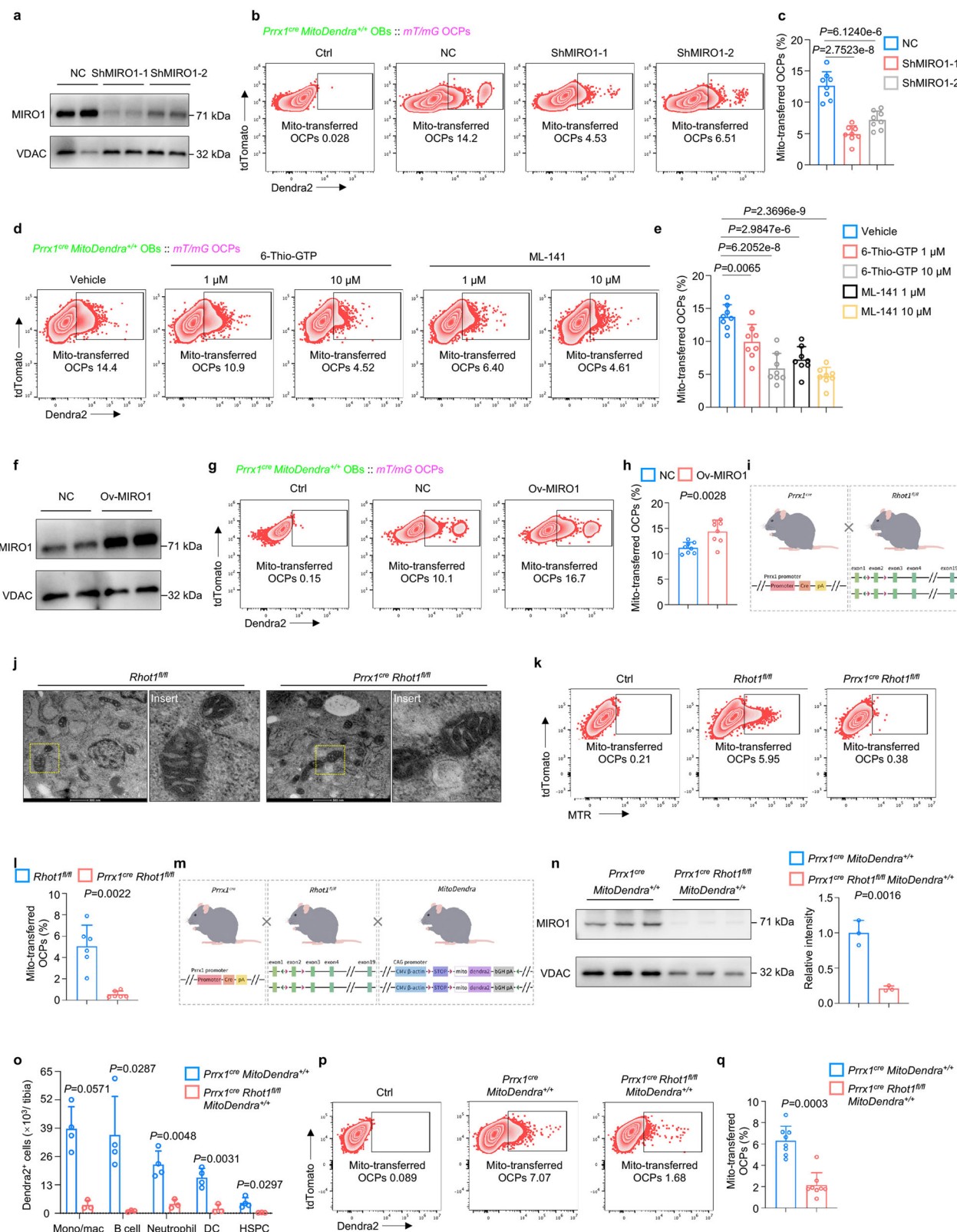

Given that MFN2 has also been reported to regulate the mitochondrial transfer between osteocytes[44], we also downregulated and upregulated MFN2 in OBs, respectively (Supplementary Fig. 4b, e) and found that mitochondrial transfer was not significantly altered (Supplementary Fig. 4c, d, f, g), implying that MFN2 does not participate in regulating the mitochondrial transfer process from osteolineage cells to osteoclastic lineage cells.

Furthermore, we established osteolineage MIRO1 knockout mice by crossing *Prrx1cre* mice with *Rhot1fl/fl* mice (Fig. 4i and Supplementary Fig. 4h). Transmission electron microscopy (TEM) detected a normal crista ultrastructure of mitochondria in the OBs extracted from the *Prrx1cre Rhot1fl/fl* mice (Fig. 4j). Additionally, mitochondria mass (Supplementary Fig. 4i, j), membrane potential (Supplementary Fig. 4k, l) and ROS (Supplementary Fig. 4m, n) were not altered compared to

**Fig. 4 | MIRO1 mediates the intercellular mitochondrial transfer. a** Western blot of OBs transduced with *Rhot1* shRNA lentiviral particles demonstrating down-regulation of MIRO1. **b**, **c** Representative images of flow cytometry (**b**) and analysis (**c**) showing decreased transfer of Dendra2-positive mitochondria from OBs to OCPs after MIRO1 downregulation (*n* = 8 per group). **d**, **e** Representative images of flow cytometry (**d**) and analysis (**e**) showing decreased transfer of Dendra2-positive mitochondria from OBs to OCPs after pharmacological inhibitor stimulation (*n* = 8 per group). **f** Western blot of OBs transduced with *Rhot1* shRNA lentiviral particle demonstrating upregulation of MIRO1. **g**, **h** Representative images of flow cyto-metry (**g**) and analysis (**h**) showing increased transfer of Dendra2-positive mito-chondria from OBs to OCPs after MIRO1 upregulation (*n* = 8 per group). **i** Schematic of generation of the *Prrx1*$^{cre}$ *Rhot1*$^{fl/fl}$ transgenic mouse line. This figure was created by P.D. and cartoonized by Mr. Zihao Li. **j** TEM showing crista ultrastructure of OB mitochondria in *Rhot1*$^{fl/fl}$ and *Prrx1*$^{cre}$ *Rhot1*$^{fl/fl}$ mice. Scale bar, 500 nm. **k**, **l** Representative images of flow cytometry (**k**) and analysis (**l**) showing decreased transfer of MTR-labeled mitochondria from OBs to OCPs after MIRO1 deficiency

(*n* = 6 per group). **m** Schematic of the generation of the *Prrx1*$^{cre}$ *Rhot1*$^{fl/fl}$ *MitoDendra*$^{+/+}$ transgenic mouse line. This figure was created by P.D. and cartoo-nized by Mr. Zihao Li. **n** Western blot and quantification demonstrating the knockout of Miro1 in osteoprogenitors in vivo (*n* = 3 per group). **o** Flow cytometry analysis showing decreased mitochondrial transfer from osteolineage cells to other lineage cells in vivo (*n* = 4 in *Prrx1*$^{cre}$ *MitoDendra*$^{+/+}$ group and *n* = 3 in *Prrx1*$^{cre}$ *Rhot1*$^{fl/fl}$ *MitoDendra*$^{+/+}$ group). **p**, **q** Representative images of flow cytometry (**p**) and analysis (**q**) showing decreased transfer of Dendra2-positive mitochondria from OBs to OCPs after MIRO1 deficiency (*n* = 8 per group). Data are presented as the mean ± s.d., with biologically individual data points shown. *P* values were determined by ordinary one-way ANOVA test with Tukey's multiple comparisons (**c**, **e**), unpaired two-tailed Student's *t* test (**h**, **n**, neutrophils and DCs of **o**), unpaired two-tailed Student's *t* test with Welch's correction (B cells and HSPCs of **o**), two-tailed Mann–Whitney *U* test (**l**, mono/mac of **o**, **q**). Source data are provided as a Source Data file.

those of the *Rhot1*$^{fl/fl}$ mice, indicating that osteolineage MIRO1 defi-ciency does not affect their own mitochondria functions, which is also consistent with a previous report observed in neurons[41]. However, when these cells lacking MIRO1 were cocultured with OCPs, the mitochondrial transfer process was significantly impaired (Fig. 4k, l). To specifically examine the role of MIRO1 in mitochondrial transfer of osteolineage cells, we crossed osteolineage MIRO1 knockout mice with the mitochondria reporter mice to generate a mouse (*Prrx1*$^{cre}$ *Rhot1*$^{fl/fl}$ *MitoDendra*$^{+/+}$ reporter mice) in which MIRO1 was knocked-out and mitochondria were fluorescently labeled selectively in osteolineage cells (Fig. 4m, n). There was no difference in the number of myeloid cells (including monocytes/macrophages, neutrophils and DCs), B220$^+$ lymphoid cells or HSPCs between *Prrx1*$^{cre}$ *MitoDendra*$^{+/+}$ and *Prrx1*$^{cre}$ *Rhot1*$^{fl/fl}$ *MitoDendra*$^{+/+}$ mice (Supplementary Fig. 4o). The impairment of mitochondrial transfer to monocytes/macrophages, B cells, neu-trophils, DCs and HSPCs in bone marrow was observed in the *Prrx1*$^{cre}$ *Rhot1*$^{fl/fl}$ *MitoDendra*$^{+/+}$ reporter mice compared to wild-type reporter mice (Fig. 4o). In addition, the mitochondrial transfer ability of OBs extracted from *Prrx1*$^{cre}$ *Rhot1*$^{fl/fl}$ *MitoDendra*$^{+/+}$ reporter mice were impaired in vitro (Fig. 4p, q). Collectively, these data demonstrated that MIRO1 mediates mitochondrial transfer from osteolineage cells to osteoclastic lineage cells.

## Impaired mitochondrial transfer results in osteoporosis

To determine whether MIRO1-mediated mitochondrial transfer reg-ulates skeletal metabolic homeostasis, we examined the bone pheno-types of the osteolineage MIRO1 deficient mouse model. Whole mount skeletal staining revealed that there was no apparent difference in craniofacial or long bones between the neonatal *Rhot1*$^{fl/fl}$ and *Prrx1*$^{cre}$ *Rhot1*$^{fl/fl}$ mice at P0 (Supplementary Fig. 5a), indicating that osteoline-age mitochondrial transfer is not involved in the regulation of embryonic skeletal development. However, when mice grew up to 8 weeks, significant bone loss was observed in the appendicular ske-leton in the *Prrx1*$^{cre}$ *Rhot1*$^{fl/fl}$ mice, as confirmed by a significant decrease in femur bone mineral density (BMD), bone volume fraction (BV/TV), trabecular number (Tb.N), trabecular thickness (Tb.Th), as well as greater trabecular separation (Tb.Sp) in the *Prrx1*$^{cre}$ *Rhot1*$^{fl/fl}$ mice (Fig. 5a, b). In addition, there was a decline in cortical bone mass with decreased cortical thickness (Ct.Th) and increased cortical porosity (Ct.Po) (Fig. 5a, c). Intriguingly, microcomputed tomography (μCT) analysis of the axial skeleton also revealed significant bone loss in vertebral bodies (Supplementary Fig. 5b–d). As Prrx1 mainly marks osteoprogenitors and there were no developmental defects after MIRO1 deficiency in the *Prrx1*$^{cre}$ *Rhot1*$^{fl/fl}$ mice, we further utilized tamoxifen-inducible osteoblastic cells expressing *Col1a1*$^{cre/ERT2}$ to induce MIRO1 deletion at 6 weeks of age for 4 weeks (Supplementary Fig. 6a). Consistent with the bone loss phenotype observed in the *Prrx1*$^{cre}$ *Rhot1*$^{fl/fl}$ mice, there was a remarkable bone loss of trabecular

bone (Supplementary Fig. 6b, c) and cortical bone (Supplementary Fig. 6b, d) in the *Col1a1*$^{cre/ERT2}$ *Rhot1*$^{fl/fl}$ mice. Similarly, bone loss of vertebral bodies in the *Col1a1*$^{cre/ERT2}$ *Rhot1*$^{fl/fl}$ mice was also observed (Supplementary Fig. 6e, f). Together, these data indicated that impaired mitochondrial transfer in osteolineage cells results in pri-mary osteoporosis.

To clarify the relative contributions of osteoblast activity and osteoclast activity to the bone loss observed in the *Prrx1*$^{cre}$ *Rhot1*$^{fl/fl}$ mice, we first analyzed osteoprogenitors (CD11b$^-$CD45$^-$CD29$^+$Sca1$^+$)[45] and found that their numbers were not significantly affected by MIRO1 deficiency (Supplementary Fig. 5e, f). We further sorted these cells and determined their osteogenesis potential in vitro. The deletion of MIRO1 did not alter osteogenic differentiation, as indicated by unaf-fected alkaline phosphatase (ALP) staining (Supplementary Fig. 5g) and alizarin red S (ARS) staining (Supplementary Fig. 5h). We then performed histomorphometry analysis of the *Rhot1*$^{fl/fl}$ and *Prrx1*$^{cre}$ *Rhot1*$^{fl/fl}$ mice to evaluate static and dynamic parameters of bone for-mation and bone resorption. Consistent with the μCT data, histo-morphometric analysis showed that the *Prrx1*$^{cre}$ *Rhot1*$^{fl/fl}$ mice had a significant decrease in both BV/TV and Tb.N (Fig. 5d, e). Intriguingly, the number of osteoblasts (N.Ob/BS) and osteoid-covered surface (OS/BS) were not significantly changed in the *Prrx1*$^{cre}$ *Rhot1*$^{fl/fl}$ mice com-pared to the *Rhot1*$^{fl/fl}$ mice (Fig. 5d, f). Consistently, the mineral surface (MS/BS), mineral apposition rate (MAR), and bone formation rate (BFR/BS) were not significantly changed in the *Prrx1*$^{cre}$ *Rhot1*$^{fl/fl}$ mice (Fig. 5g, h). The serum bone formation index, procollagen type 1 N-terminal propeptide (P1NP) and BALP were not significantly altered after MIRO1 deficiency (Fig. 5i). Furthermore, osteoblasts were extracted from P0 pups and osteogenesis was assessed. ALP and ARS staining revealed no difference between the *Rhot1*$^{fl/fl}$ and *Prrx1*$^{cre}$ *Rhot1*$^{fl/fl}$ mice (Fig. 5j, k), and the mRNA levels of osteogenic markers, including *Col1a1*, *Spp1*, *Sp7* and *Runx2*, were not significantly changed (Supplementary Fig. 5i). We also confirmed these phenotypes in the *Col1a1*$^{cre/ERT2}$ *Rhot1*$^{fl/fl}$ mice with no alterations in bone formation, as revealed by static (Supplementary Fig. 6g–i) and dynamic histomor-phometric analysis (Supplementary Fig. 6j, k) and serum P1NP con-centrations (Supplementary Fig. 6l). Consistent with unaffected mitochondria functions after MIRO1 deficiency in osteolineage cells, we demonstrated that osteolinege MIRO1 deletion does not affect osteoblast activities.

We then performed tartrate-resistant acid phosphatase (TRAP) staining to examine osteoclast activity in the *Prrx1*$^{cre}$ *Rhot1*$^{fl/fl}$ mice. Interestingly, we found an increase of osteoclasts along the surface of trabecular bone in the *Prrx1*$^{cre}$ *Rhot1*$^{fl/fl}$ mice (Fig. 5l) with increased osteoclast number (N.Oc/BS) and osteoclast surface (Oc.S/BS) (Fig. 5m). The serum bone resorption index, collagen type I c-telopeptide (CTX) and TRAcP-5b were also significantly augmented in the *Prrx1*$^{cre}$ *Rhot1*$^{fl/fl}$ mice (Fig. 5n). Similarly, enhanced osteoclast

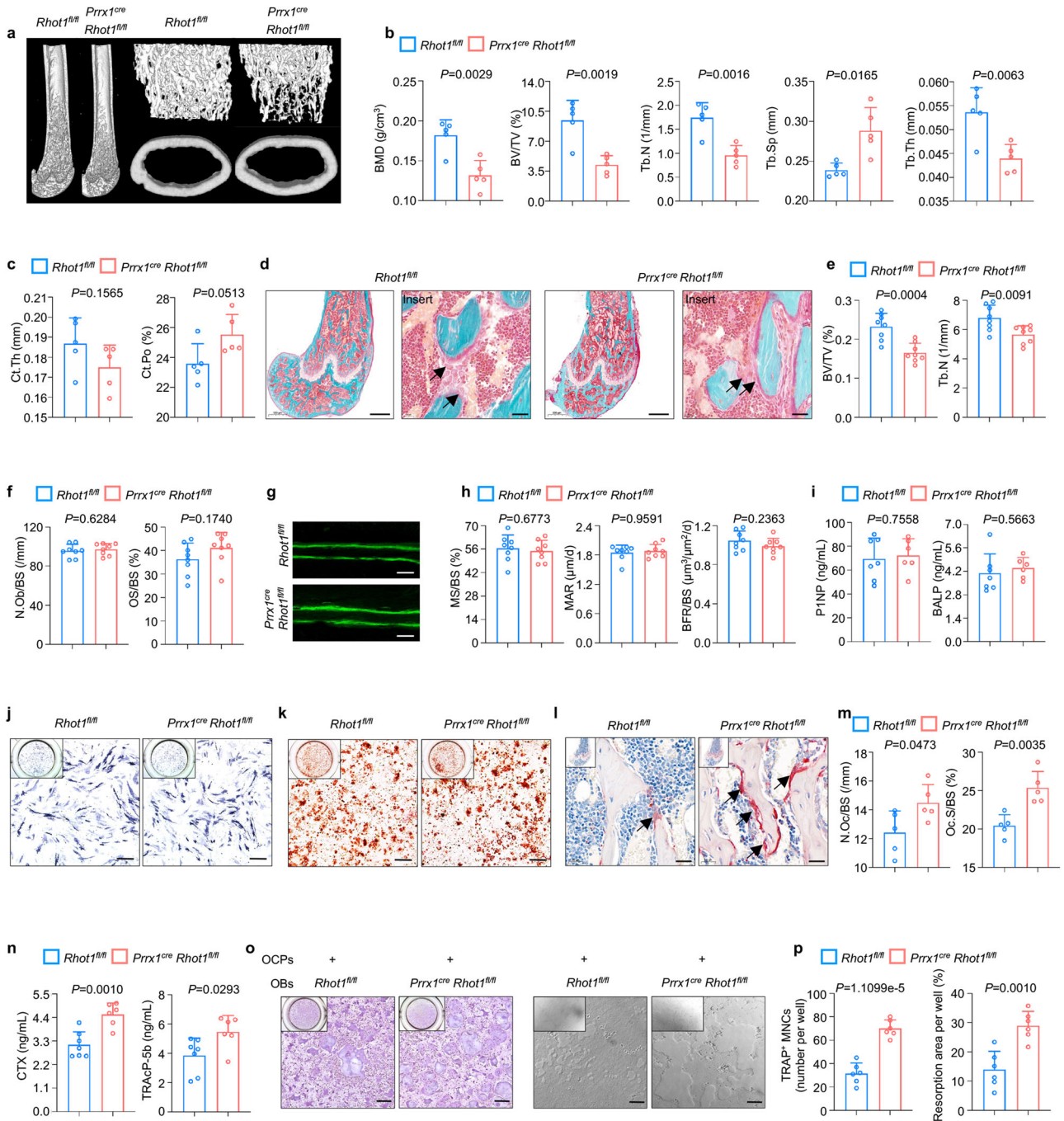

**Fig. 5 | Impaired mitochondrial transfer results in osteoporosis.**
**a–c** Representative µCT reconstructed images of male *Rhot1^fl/fl^* and *Prrx1^cre^ Rhot1^fl/fl^* mouse femurs at 8 weeks (**a**) and trabecular microstructural parameters (BMD, BV/TV, Tb.N, Tb.Sp and Tb.Th) (**b**) and cortical microstructural parameters (Ct.Th and Ct.Po) (**c**) derived from µCT analysis (*n* = 5 per group). **d–f** Goldner trichrome staining of male *Rhot1^fl/fl^* and *Prrx1^cre^ Rhot1^fl/fl^* mouse femurs at 8 weeks (**d**) and histomorphometry analysis of BV/TV, Tb.N, N.Ob/BS and OS/BS (**f**) (*n* = 8 per group). Scale bar, 500 and 50 µm. **g, h** Representative images of calcein double labeling of the mineral layers of male *Rhot1^fl/fl^* and *Prrx1^cre^ Rhot1^fl/fl^* mice femur at 8 weeks (**g**) and histomorphometric analysis of MS/BS, MAR and BFR/BS (**h**) (*n* = 8 per group). Scale bar, 50 µm. **i** ELISAs of the concentration of bone formation index P1NP and BALP in the serum (*n* = 7 in *Rhot1^fl/fl^* group and *n* = 6 in *Prrx1^cre^ Rhot1^fl/fl^* group). **j, k**, ALP (**j**) and

ARS (**k**) staining of osteogenesis in vitro from *Rhot1^fl/fl^* and *Prrx1^cre^ Rhot1^fl/fl^* mice. Scale bar, 250 µm. **l, m** TRAP staining of male *Rhot1^fl/fl^* and *Prrx1^cre^ Rhot1^fl/fl^* mice femur at 8 weeks (**l**) and histomorphometric analysis of N.Oc/BS and Oc.S/BS (**m**) (*n* = 5 per group). Scale bar, 50 µm. **n** ELISAs of the concentration of bone resorption index CTX and TRAcP-5b in the serum (*n* = 7 in *Rhot1^fl/fl^* group and *n* = 6 in *Prrx1^cre^ Rhot1^fl/fl^* group). **o, p** In vitro osteoblast-osteoclast coculture model to induce osteoclastogenesis and bone resorption (**o**) and quantification of TRAP positive cells (nucleus > 3) per well and resorption area per well (**p**) (*n* = 6 per group). Scale bar, 250 µm. Data are presented as the mean ± s.d., with biologically individual data points shown. *P* values were determined by unpaired two-tailed Student's *t*-test with Welch's correction (Tb.Sp of **b**, **m**), unpaired two-tailed Student's *t* test (**b**, **c**, **e**, **f**, **h**, **i**, **n**, **p**), and two-tailed Mann–Whitney *U* test (MAR of **h**). Source data are provided as a Source Data file.

activities were observed in the *Col1a1^cre/ERT2^ Rhot1^fl/fl^* mice with elevated CTX in the serum (Supplementary Fig. 6l) and increased osteoclasts (Supplementary Fig. 6m, n). In addition, after coculture with MIRO1 deficient osteoblasts from the *Prrx1^cre^ Rhot1^fl/fl^* mice, osteoclast

differentiation and resorption were substantially promoted compared to those with coculture with wild-type osteoblasts (Fig. 5o, p). Together, these data demonstrated that osteolineage MIRO1 deficiency induces osteoclastic bone loss. However, serum concentrations of

RANKL and osteoprotegerin (OPG) and the ratio of RANKL/OPG (Supplementary Fig. 5j) as well as the expression of *Tnfrsf11* and *Tnfrsf11b* in osteoblasts (Supplementary Fig. 5i) showed no significant difference between the *Rhot1^{fl/fl}* and *Prrx1^{cre} Rhot1^{fl/fl}* mice, ruling out the possibility that soluble factors secreted by osteolineage cells tune the activity of osteoclasts. These results indicated that impaired mitochondrial transfer from osteolineage cells to osteoclastic lineage cells results in osteoclastic bone loss.

Osteocytes, as the terminally differentiated cells of osteolineage cells, have been reported to regulate osteoclast activities[46] and transfer mitochondria within their dendritic network[44]. We employed *Dmp1^{cre} Rhot1^{fl/fl}* mice (Supplementary Fig. 7a), in which MIRO1 was deleted in osteocytes, and found that there was no significant difference in trabecular and cortical bone mass between the *Rhot1^{fl/fl}* and *Dmp1^{cre} Rhot1^{fl/fl}* mice (Supplementary Fig. 7b–d). The bone mass of vertebral bodies revealed no difference (Supplementary Fig. 7e, f), either. Further, the serum bone formation and resorption index (Supplementary Fig. 7g), as well as the number and surface area of osteoclasts (Supplementary Fig. 7h, i), were not significantly altered after MIRO1 deficiency. These data revealed that osteocytic MIRO1 does not regulate bone homeostasis. Consistent with the much lower mitochondrial transfer to other lineage cells in the *Dmp1^{cre} MitoDendra^{+/+}* reporter mice compared to the *Prrx1^{cre} MitoDendra^{+/+}* reporter mice and the *Col1a1^{cre/ERT2} MitoDendra^{+/+}* reporter mice (Fig. 1j), we further inferred that mitochondrial transfer from osteolineage cells regulating bone homeostasis is dependent on the number of mitochondrial transferred.

We also validated the bone phenotype of the *Lyz2^{cre} Rhot1^{fl/fl}* mice (Supplementary Fig. 8a). MIRO1 deficiency in osteoclastic lineage cells did not affect bone mass, as confirmed in femur trabecular (Supplementary Fig. 8b, c), cortical bone (Supplementary Fig. 8b, d) and vertebral bodies (Supplementary Fig. 8e, f). There was no significant difference in the serum bone formation and resorption index (Supplementary Fig. 8g) or the osteoclast number and surface area (Supplementary Fig. 8h, i) between the *Rhot1^{fl/fl}* and *Lyz2^{cre} Rhot1^{fl/fl}* mice. Additionally, in vitro osteoclast differentiation showed no difference between the *Rhot1^{fl/fl}* and *Lyz2^{cre} Rhot1^{fl/fl}* mice (Supplementary Fig. 8j, k). Together, these data demonstrated that osteoclastic MIRO1 does not regulate bone homeostasis, in accordance with the phenomenon that mitochondrial transfer does not occur from the osteoclastic lineage to osteolineage cells. Collectively, we concluded that MIRO1 acts primarily in osteolineage cells to transfer mitochondria to osteoclastic lineage cells and therefore orchestrates bone resorption.

## Osteolineage mitochondria alter osteoclastic GSH metabolism

Considering that mtD2^{pos} OCPs had altered cell activities and metabolism, we reasoned that osteolineage cells derived mitochondria exerted effects via metabolic processes. We first performed mitochondria transplantation, as confirmed by about 13% mitochondria intake compared to that in the vehicle-treated group (Fig. 6a). Transplanted mitochondria exerted inhibitory effects on osteoclast differentiation (Fig. 6b, c) with profoundly decreased expression of osteoclast-related signature genes, including *Acp5*, *Ctsk*, *Nfatc1*, *Atp6v0d2*, *Ocstamp* and *Dcstamp* (Fig. 6d). These data validated that osteolineage cell-derived mitochondria exert inhibitory effects on osteoclast activities. To further investigate the fate of osteolineage cells-derived mitochondria following their transfer into myeloid cells, we stained the MitoTracker to examine whether there is colocalization of Dendra2 signal in the mitochondria of recipient cells. Intriguingly, there was significant colocalization of Dendra2 signal in the mitochondria of recipient cells (Fig. 6e, f). We inferred that transferred mitochondria are active and functional and exert its functions by interactions with mitochondria network of recipient cells. We then transplanted osteolineage cell-derived mitochondria into osteoclastic lineage cells at different stages and performed RNA sequencing and

untargeted metabolomics, as illustrated in Fig. 6g. Partial least squares-discriminant analysis (PLS-DA) demonstrated a reliable separation in the NEG between the BMMs and BMMs^{Mito} (BMMs transplanted with osteolineage cell-derived mitochondria) (Fig. 6h and Supplementary Data 1). Hierarchical clustering and heatmap analysis revealed that BMMs and BMMs^{Mito} were grouped into distinct metabolite clusters, indicating a marked difference in their metabolic signatures (Fig. 6i). The top ten KEGG pathway enrichments of these significantly changed metabolites further demonstrated that the ferroptosis pathway was most significantly affected pathway (Fig. 6j). Along with RNA sequencing, we performed integrative analysis, and a Venn plot showed that a total of 23 KEGG pathways were obtained repeatedly from both metabolomics and transcriptomics data-based pathway analysis (Fig. 6k). Furthermore, integrated enrichment analysis based on the KEGG pathway revealed that transplanted mitochondria perturbed glutathione metabolism and the ferroptosis pathway (Fig. 6l). Consistently, the top seven enriched KEGG pathways significantly altered metabolites between the mOCs and mOCs^{Mito} (mOCs transplanted with osteolineage cell-derived mitochondria) also revealed that the ferroptosis, cysteine and methionine metabolism, and glutathione metabolism pathways were significantly affected (Supplementary Fig. 9a and Supplementary Data 2). Similarly, these pathways were identified as the most relevant pathways affected by mitochondria transplantation, as revealed by integrated enrichment analysis (Supplementary Fig. 9b). In addition, KEGG pathway enrichment of significantly altered metabolites between the OCPs and OCPs^{Mito} (OCPs transplanted with osteolineage cell-derived mitochondria) demonstrated that the citrate cycle (TCA cycle) was significantly affected (Supplementary Fig. 9c and Supplementary Data 3), and Seahorse metabolic rate analysis confirmed that transplantation of osteolineage cell-derived mitochondria led to significant reductions in the oxygen consumption rate (Supplementary Fig. 9d). Collectively, these data disclosed that transplantation of osteolineage cell-derived mitochondria results in the altered metabolism of osteoclastic lineage cells.

As GSH is the key regulator of ferroptosis[47,48], and our integrated enrichment analysis revealed that both the glutathione metabolism and ferroptosis pathways were enriched, we hypothesized that osteolineage cell-derived mitochondria altered the glutathione metabolism of osteoclastic lineage cells, leading to ferroptosis (Fig. 6m). To test this hypothesis, we first measured GSH, and the amounts of GSH were significantly reduced after mitochondria transplantation (Fig. 6n). We then examined the expression of GSH metabolism-related genes, in which the mRNA levels of *Gss*, *Gsr*, *Gclc*, *Gclm*, *Gls*, and *Gpx4* were significantly decreased when mitochondria were transplanted (Fig. 6o). These data indicated that osteolineage cell-derived mitochondria inhibit GSH metabolism, especially GSH synthesis in osteoclastic lineage cells. In addition, we found that mitochondria transplantation decreased cell viability (Fig. 6p), which was consistent with the RNA sequencing results showing impaired cell cycle and DNA replication after mitochondria transplantation (Supplementary Fig. 9e–g), and GSH depletion by erastin and buthionine sulphoximine (BSO) exerted similar effects (Fig. 6p). Further, when mitochondria transplanted cells received GSH repletion with either GSH or N-acetylcysteine (NAC), their cell viability retrieved (Fig. 6p). Additionally, treatment with the ferroptosis inhibitor ferrostatin-1 (Fer-1) rescued the decreased cell viability of mitochondria-transplanted cells (Fig. 6p). To further confirm the occurrence of ferroptosis, we performed Liperfluo staining, FerroOrange staining, TUNEL assays and transmission electron microscopy (TEM) in mitochondria-transplanted cells with or without GSH repletion or Fer-1 rescue. Consistent with the CCK-8 assay results, the TUNEL assay confirmed that mitochondria transplantation-induced cell death could be significantly rescued by GSH repletion with GSH or Fer-1 treatment (Supplementary Fig. 9h). Moreover, GSH depletion with BSO induced similar cell death (Supplementary Fig. 9h). Liperfluo staining showed

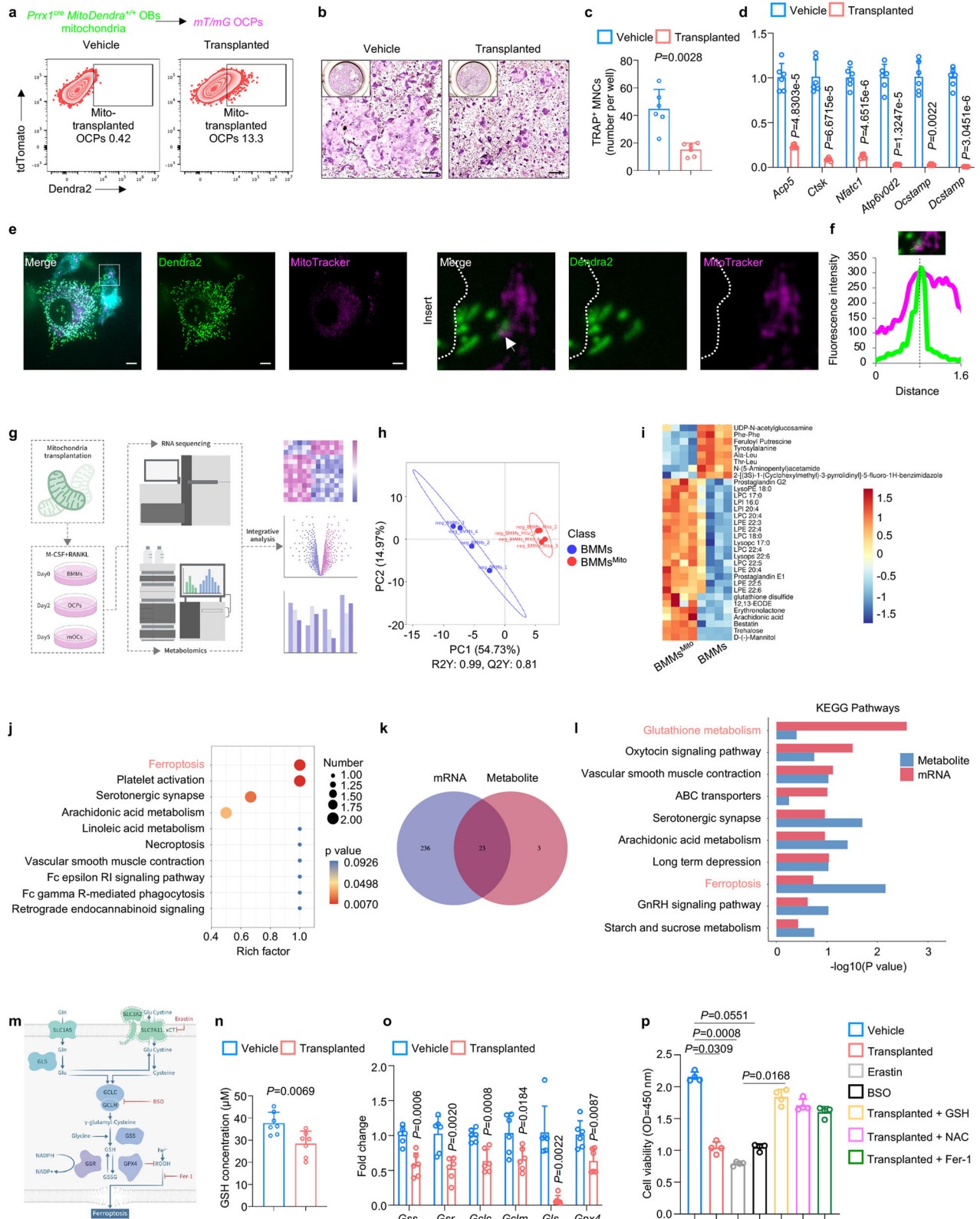

obvious lipid peroxidation in response to mitochondria transplant and GSH depletion, and this effect was rescued by GSH repletion or Fer-1 treatment (Supplementary Fig. 9h). FerroOrange is an $Fe^{2+}$-specific probe, whose fluorescence increased drastically after mitochondria transplant and GSH depletion, and this change could be reversed by GSH repletion or Fer-1 treatment (Supplementary Fig. 9h). By TEM, we

also observed shrunken mitochondria with ruptured membranes after mitochondria transplantation, and GSH repletion or Fer-1 treatment partially restored the mitochondrial structure (Supplementary Fig. 9h). Taken together, these data demonstrated that osteolineage cell-derived mitochondria inhibit osteoclast activities by inhibiting GSH metabolism, thus leading to ferroptosis.

**Fig. 6 | Osteolineage mitochondria alter osteoclastic GSH metabolism.**
**a** Representative images of flow cytometry showing the uptake of Dendra2-positive mitochondria into OCPs after transplantation. **b, c** TRAP staining of in vitro osteoclastogenesis (**b**) after mitochondria transplantation and quantitative analysis (**c**) of TRAP-positive cells (nucleus > 3) per well (*n* = 6 per group). Scale bar, 250 μm. **d** qPCR analysis of the expression of osteoclast signature genes at the mRNA level after mitochondria transplantation (*n* = 6 per group). **e, f** Representative images (**e**) and fluorescence intensity analysis (**f**) showing the interaction between mitochondria derived from OBs and mitochondria in BMMs. Scale bar, 25 μm.
**g** Schematic showing the experimental design of mitochondria transplantation and subsequent sequencing. This figure was created by P.D. and cartoonized by Mr. Zihao Li. **h**, PLSDA analysis of metabolomics of BMMs and BMMs^Mito (BMMs transplanted with osteolineage cell-derived mitochondria). **i, j** Heatmap of untargeted metabolomics showing differentially modulated metabolites (**i**) and the top ten enriched KEGG pathways showing that the ferroptosis pathway was most significantly affected (**j**). **k, l** Venn plot of integrative analysis of RNA sequencing and

metabolomics demonstrating that a total of 23 KEGG pathways were obtained (**k**) and integrated enrichment analysis based on the KEGG pathway showing the enrichment of glutathione metabolism and ferroptosis pathway (**l**). **m** Schematic illustrating the mechanism of glutathione metabolism and induction of ferroptosis. **n, o** Measurement of GSH (**n**) and qPCR of the expression of glutathione metabolism-related genes at the mRNA level (**o**) after mitochondria transplantation (*n* = 7 per group). **p** Cell viability of OCPs treated with mitochondria transplantation, glutathione depletion (Erastin and BSO), mitochondria transplantation with glutathione repletion (GSH and NAC) and ferroptosis inhibitors (Fer-1) measured by the CCK-8 test (*n* = 4 per group). Data are presented as the mean ± s.d., with biologically individual data points shown. *P* values were determined by unpaired two-tailed Student's *t* test (**n, o**), two-tailed Mann–Whitney *U* test (*Ocstamp* of **d**, *Gls* and *Gpx4* of **o**), unpaired two-tailed Student's *t* test with Welch's correction (**c, d**), hypergeometric test (**j**) and non-parametric ANOVA with Dunn's multiple comparisons test (**p**). Source data are provided as a Source Data file.

## Osteolineage mitochondrial transfer regulates GIOP

Glucocorticoid (GC) medications, such as prednisone or cortisone, are commonly prescribed for various inflammatory and autoimmune conditions[49], and the main cause of GIOP. Besides primary osteoporosis, GIOP represents the most common type of secondary osteoporosis[50,51], which is also closely linked with energy metabolism[52]. To investigate the role of mitochondrial transfer from osteolineage cells to myeloid cells in GIOP, we first compared these reporter mice with or without GC drinking water treatment for 8 weeks[53] (Fig. 7a). While GC treatment significantly promoted the expansion of HSPCs and myeloid cells at the expense of decreased B220+ lymphoid cells (Supplementary Fig. 10a, b), GC treatment significantly inhibited the expression of MIRO1 in osteolineage cells (Supplementary Fig. 10c), and an impaired mitochondrial transfer from osteolineage cells to myeloid cells, including monocytes/macrophages, neutrophils, DCs and lymphoid cells, including B cells was observed (Fig. 7a). However, GC treatment did not significantly impair mitochondrial transfer to HSPCs (Fig. 7a). Meanwhile, we also investigated whether mitochondrial transfer regulated other pathophysiological bone diseases using ovariectomized mice model and bone repair model. And we uncovered that the mitochondrial transfer from osteolineage cells to myeloid cells was also impaired in the ovariectomized mice (Supplementary Fig. 10d). Meanwhile, during bone repair following bone injury, decreased mitochondrial transfer from osteolineage cell to myeloid cells was also observed (Supplementary Fig. 10e). Together, all these data confirmed the important role of mitochondrial transfer in regulation of bone homeostasis.

We then pretreated OBs with different concentrations of GC and then cocultured with OCPs. 10 nM and 100 nM GC pretreatment did not significantly alter OB proliferation (Supplementary Fig. 10f) or mitochondrial biogenesis (Supplementary Fig. 10g, h). However, the MIRO1 expression in OBs was significantly inhibited after GC pretreatment (Supplementary Fig. 10i). Flow cytometry analysis revealed a significant decrease in mitochondrial transfer with the increasing GC concentration, in which when the GC concentration came up to 100 nM, a drastic decrease of mitochondrial transfer was observed, and 1000 nM GC treatment, which reflects the pathological concentrations[53], inhibited the most mitochondrial transfer (Fig. 7b, c). These data indicated that GC treatment impairs the mitochondrial transfer from osteolineage to myeloid cells.

In addition to suppress bone formation, GC treatment promotes bone resorption during the initial phase[54,55] by regulating the supply and lifespan of osteoclasts[56]. While there were studies investigating the mechanism from the aspect of bone formation and resorption, there were no efficient therapies treating GIOP. Targeting the crosstalk between osteoblasts and osteoclasts could be a promising strategy for the treatment of GIOP. As GC treatment impaired the mitochondrial

transfer from osteolineage cells to osteoclastic lineage cells in vivo and in vitro and osteolineage cell-derived mitochondria altered the GSH metabolism of osteoclastic lineage cells, we then explored whether mitochondrial transfer from osteolineage cells to myeloid cells participates in the regulation of GC-induced osteoclastogenesis. We demonstrated that extra addition of osteolineage cell-derived mitochondria or depletion of GSH with BSO under the GC treatment could reverse GC-induced changes in osteoclast activity. Specifically, after coculture with GC-pretreated OBs, myeloid cell commitments toward osteoclastogenesis were significantly promoted (Fig. 7d, e), and this promoting effect was impaired by the transplantation of osteolineage cells derived mitochondria (Fig. 7d, e), or depleting GSH with BSO (Fig. 7d, e), indicating that osteolineage cells derived mitochondria regulate GC-induced osteoclast activity via altering the GSH metabolism. Together, these data demonstrated that mitochondrial transfer from osteolineage cells to myeloid cells contributes to GC-induced osteoclastogenesis.

We then further examined whether GSH depletion by BSO could alleviate the progression of GIOP. To analyze the effects of BSO on GC-induced bone loss in vivo, we subjected the mice to GC drinking water treatment for 8 weeks, and BSO (8 mmol/kg) was injected intraperitoneally into mice at 4-day intervals for 8 weeks. After 8 weeks, the BSO-administered mice (GC + BSO group) were normal in appearance with a body size similar to the vehicle-administered mice (GC + vehicle group), and BSO had no effect on body weight as previously reported[57]. Also, there was no significant difference of the number of myeloid cells (including monocytes/macrophages, neutrophils and DCs), B220+ lymphoid cells and HSPCs between GC + Vehicle and GC + BSO group (Supplementary Fig. 10a, b). We firstly verified the bone loss phenotype induced by GC (GC + vehicle group) compared to that of the control group (Ctrl group, mice without GC drinking water treatment) (Fig. 7f–k). Intriguingly, We found that BSO administration significantly attenuated the bone loss caused by GC treatment, as confirmed by the increased BMD, BV/TV, Tb.N and Tb.Th and decreased Tb.Sp of femur trabecular bone in the BSO-administered mice compared to the vehicle-administered mice (Fig. 7f, g). Similarly, there was an increase in the bone mass of cortical bone with increased Ct.Th and decreased Ct.Po in the BSO-administered mice (Fig. 7f, h). In addition, axial skeleton analysis revealed the increased bone mass in vertebral bodies after BSO treatment (Supplementary Fig. 10j, k). Moreover, histomorphometry analysis of TRAP staining showed that osteoclast activities were significantly inhibited in BSO-administered mice with decreased osteoclast number and surface (Fig. 7I, j). Together, these data demonstrated that mitochondrial transfer from osteolineage to myeloid cells regulates GSH metabolism, which may partially contribute to GIOP progression.

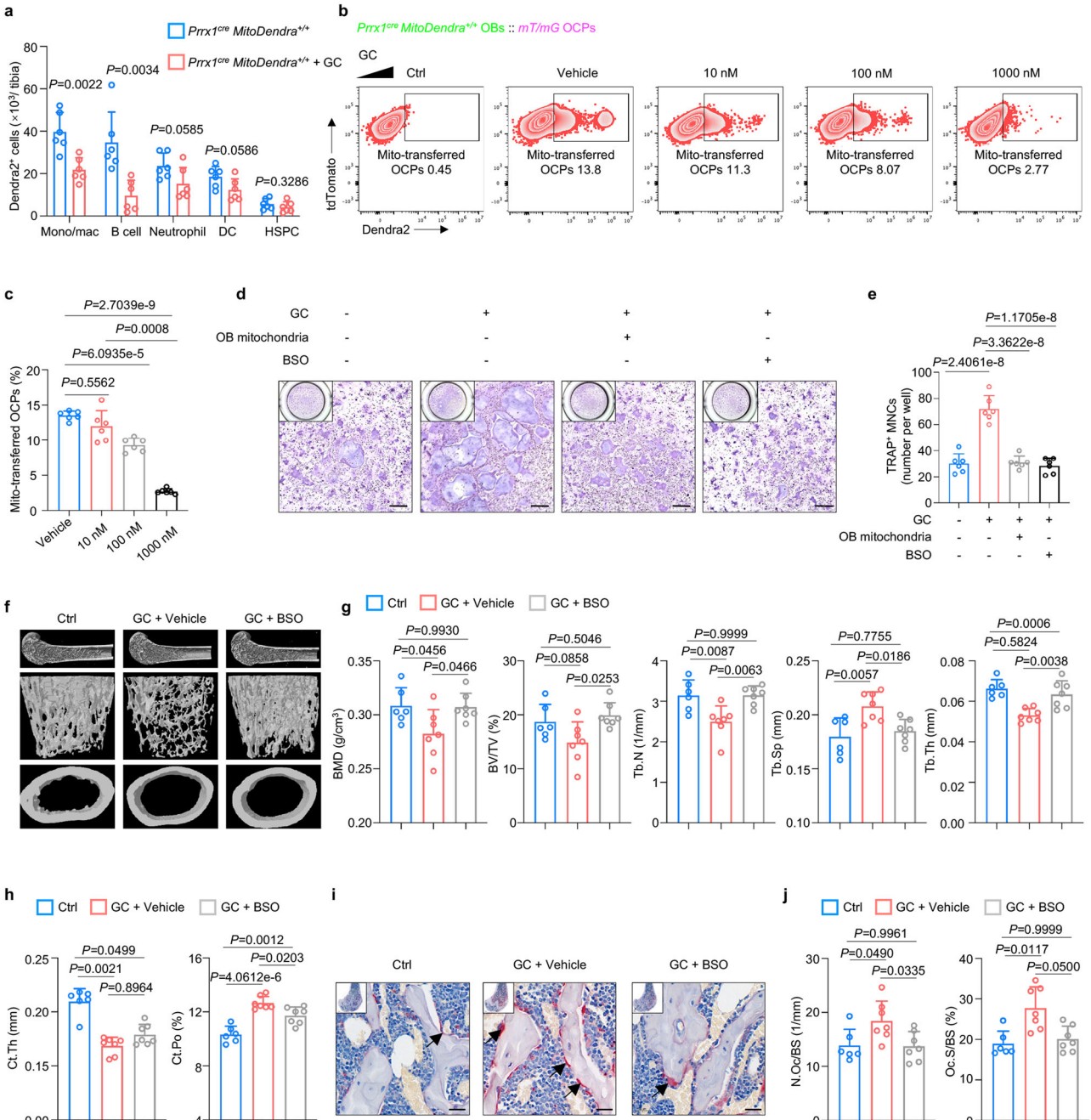

**Fig. 7 | Osteolineage mitochondrial transfer regulates GIOP. a** Flow cytometry analysis showing decreased mitochondrial transfer from osteolineage cells to other lineage cells in vivo after GC treatment ($n = 6$ per group). **b**, **c** Representative images of flow cytometry (**b**) and analysis (**c**) showing the decreased transfer of Dendra2-positive mitochondria from OBs to OCPs after GC stimulation ($n = 6$ per group). **d**, **e** In vitro osteoblast-osteoclast coculture model to induce osteoclastogenesis (**d**) and quantification of TRAP-positive cells (nucleus > 3) per well (**e**), demonstrating that GC-induced increased osteoclastogenesis was regulated by mitochondrial transfer from osteolineage to myeloid cells ($n = 3$ per group). Scale bar, 250 μm. **f**–**h** Representative μCT reconstructed images of male mouse femurs without GC treatment (Ctrl), GC-treated male mice with vehicle treatment (GC + vehicle) and GC-treated male mice with BSO treatment (GC + BSO) for 8 weeks (**f**) and trabecular microstructural parameters (BMD, BV/TV, Tb.N, Tb.Sp and Tb.Th) (**g**) and cortical

microstructural parameters (Ct.Th and Ct.Po) (**h**) derived from μCT analysis, demonstrating that GSH depletion by BSO treatment attenuated the GC induced bone loss ($n = 6$ in Ctrl group and $n = 7$ in GC + vehicle and GC + BSO group). **i**, **j** TRAP staining of male mouse femurs at 8 weeks (**i**) and histomorphometry analysis of N.Oc/BS and Oc.S/BS (**j**), demonstrating that BSO administration inhibited osteoclast activity in the GIOP mouse model ($n = 6$ in Ctrl group and $n = 7$ in GC + vehicle and GC + BSO group). Scale bar, 50 μm. Data are presented as the mean ± s.d., with biologically individual data points shown. $P$ values were determined by unpaired two-tailed Student's $t$ test (**a**), $t$wo-tailed Brown–Forsythe and Welch ANOVA tests with Dunnett's T3 multiple comparisons (**c**), two-tailed nonparametric ANOVA with Dunn's multiple comparisons test (Ct.Th of **h**, Oc.S/BS of **j**), and ordinary one-way ANOVA test with Tukey's multiple comparisons (**e**, **g**, Ct.Po of **h**, N.Oc/BS of **j**). Source data are provided as a Source Data file.

## Discussion

In this study, we found that monocytes/macrophages originating from myeloid cells represent the major recipients of osteolineage cell-derived mitochondria. We further highlighted the role of osteolineage

cells in regulating bone resorption by transferring mitochondria to myeloid cells, and inhibition of mitochondrial transfer results in osteoclastic bone loss. Transferred mitochondria inhibit GSH metabolism and subsequently lead to ferroptosis in osteoclastic lineage

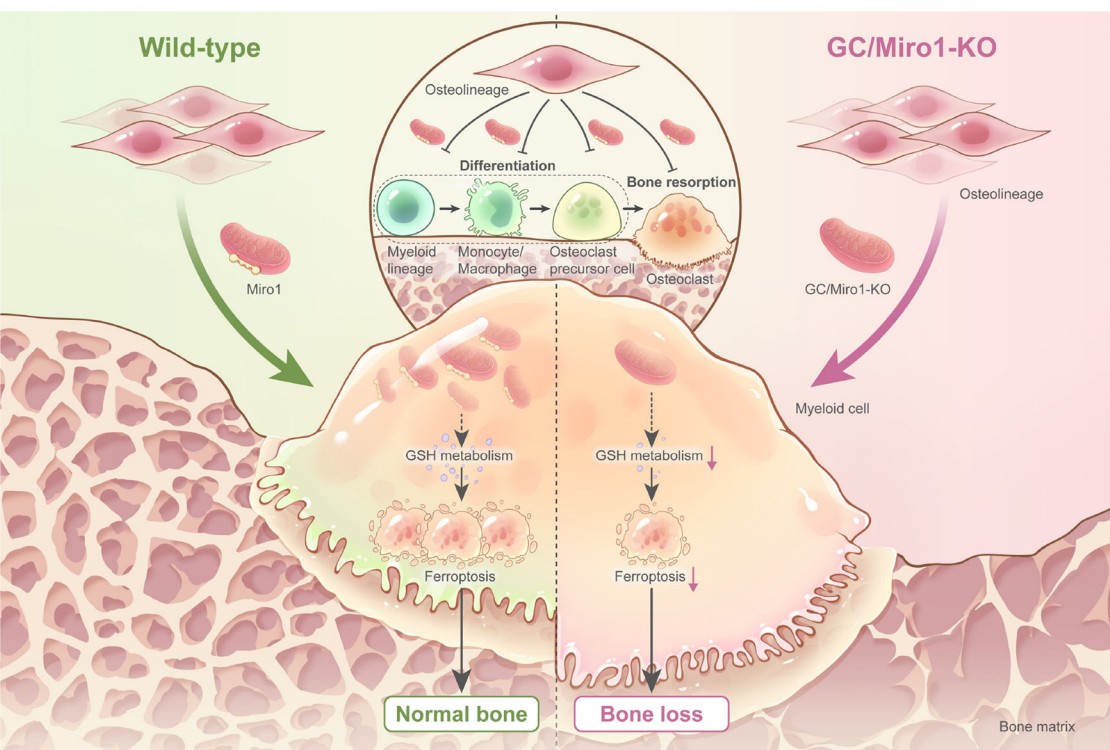

**Fig. 8 | Schematic diagram of osteolineage cell-derived mitochondria regulating myeloid cell-mediated bone resorption.** Under normal conditions, osteolineage cells transfer mitochondria to myeloid cells to inhibit their differentiation into osteoclastic lineage cells and subsequent bone resorption via GSH-mediated ferroptosis. Impairment of mitochondrial transfer by deleting MIRO1 or GC treatment alters GSH metabolism, inhibits ferroptosis of osteoclastic lineage cells and promotes osteoclast activity, thus leading to bone loss. This figure was created by P.D. and Ms. Danhong Qiu.

cells (Fig. 8). Thus, we identified a previously unappreciated mechanism to regulate bone homeostasis and contribute to the progression of GIOP.

Cell lineage commitment and interaction are not only critical for tissue/organ formation but also play indispensable roles in microenvironment maintenance. Unlike ligand-receptor crosstalk between cell interactions, cell-to-cell mitochondrial transfer represents a modulator. Different cell types gain functionally and molecularly distinct mitochondrial types as they mature during development[18,19]. Recent study has also demonstrated that cell lineage-specific mitochondrial gene expression emerges early in development which contributes to the tissue specificity[20]. And our group also reviewed the mitochondria heterogeneity between different cells[58]. And mitochondria have been increasingly recognized as the important information process system beyond the traditional role of powerhouse[29,59]. Mitochondrial transfer occurs in various tissues, such as the heart[21], neurons[28], adipocytes[22–24]. A previous study described mitochondrial transfer from adipose tissue to various nearby immune cell populations[24], indicating that mitochondrial transfer occurs in the form of a network to other cell types. However, mitochondrial transfer in the skeletal system, which hosts most lineage cells and maintains extensive communication[4], has not yet been fully deciphered. Our group previously showed that osteocytes, as the terminal osteolineage cells, transfer mitochondria to each other within their network[44]. In this study, we further established a series of osteolineage mitochondria reporter mice and found that osteolineage cells transferred mitochondria to myeloid cells, lymphoid cells and HSPCs, of which monocytes/macrophages received the most mitochondrial transfer, which is consistent with the mitochondrial transfer network observed in adipocyte tissue[24]. Previous studies have reported that mesenchymal stem cells could transfer mitochondria to T cells in vitro[60,61], and macrophages transfer mitochondria to mesenchymal stem cells in vitro[62]. However, we did not observe significant mitochondrial transfer to T cells in vivo in our osteolineage mitochondria reporter mice. We speculated that the mitochondrial transfer in vivo may be more complicated, involving continual spatiotemporal communication, and may be dependent on cell types that have different abilities and tendencies to transfer mitochondria as donor cells or receive mitochondria as recipient cells. Additionally, using the mitochondria reporter mice and their extracted cells could accurately mark mitochondria and exclude the possibility of leakage of Mito Tracker dye, which depicts the mitochondrial transfer more precisely.

Interestingly, B cells received the second largest mitochondrial transfer from osteolineage cells. Given the important role of B cells in immune responses, further studies are warranted to investigate the role of mitochondrial transfer during B cell activities. We also detected the mitochondrial transfer from osteolineage cells to HSPCs. Combined with a previous report showing that bone marrow regeneration requires HSPCs mitochondrial transfer to the mesenchymal environment[63], these data indicate that mitochondrial transfer between HSPCs and osteolineage cells may be a bilateral interactive process that is important for bone marrow regeneration. Monocytes/macrophages receive the greatest mitochondrial transfer from osteolineage cells and can differentiate into osteoclastic lineage cells. Considering that osteolineage and osteoclastic lineage cells are two main functional cell types that couple bone remodeling to maintain bone homeostasis[1], we further investigated how these two lineages of cells interact with each other. Unlike conventional views of ligand-receptor paradigm, we uncovered an unappreciated mechanism that mitochondria, as the cell information processor[29], could be transferred from osteolineage cells to osteoclastic lineage cells unidirectionally to regulate osteoclast activities. This type of mitochondrial transfer also establishes a crosstalk between bone remodeling and energy metabolism[11], as transferred mitochondria also alter GSH metabolism

and lead to ferroptosis of recipient cells. This model also provides insights that during bone remodeling, osteoclast retreat and osteoblast entry may be coupled by mitochondrial transfer, as evidenced by the fact that osteolineage cells need to directly contact osteoclastic lineage cells to perform their functions[64], and this process involves mitochondrial transfer, which inhibits osteoclast activities. Thus, osteoblasts could subsequently entry into the resorption area and build new bone.

Moreover, we demonstrated that mitochondrial transfer is mediated by MIRO1, which is consistent with previous reports showing that MIRO1 regulates mitochondria motility in neurons[40,42]. The loss of MIRO1-directed mitochondrial transfer results in osteoclastic bone loss, similarly to that observed in neuron disease caused by MIRO1 deficiency[41]. Interestingly, there was no significant bone loss in the *Dmp1cre Rhot1fl/fl* mice. Combined with the much lower mitochondrial transfer to myeloid cells observed in the *Dmp1cre MitoDendra+/+* reporter mice compared to the *Prrx1cre MitoDendra+/+* reporter mice and the *Col1a1cre/ERT2 MitoDendra+/+* reporter mice, we implied that the number of transferred mitochondria influences the bone phenotype, demonstrating the important role of mitochondrial transfer in the regulation of bone homeostasis. Further clinical studies are required in the context of human skeletal metabolic diseases to delineate the role of mitochondrial transfer.

The role of mitochondria in ferroptosis is debatable and controversial. Mitochondria can promote ferroptosis via mitochondrial ROS production, ATP generation and cellular metabolism[65–67]. However, several studies have also reported that cells depleted of mitochondria (so-called $\rho^0$ cells) are still sensitive and undergo ferroptosis[47,68]. Here, we showed that the transfer of mitochondria from another cell type could induce the ferroptosis of recipient cells via altered GSH metabolism, indicating the important role of mitochondria in ferroptosis from another aspect. A previous study demonstrated the important role of the GSH/GSSG ratio during osteoclastogenesis[57], and increasing evidence has confirmed that ferroptosis is a vital mechanism that regulates osteocyte, osteoblast and chondrocyte activities[69–72]. In this study, we defined the role of ferroptosis during osteoclastogenesis which is controlled by GSH metabolism. In fact, GSH metabolism is closely related with ferroptosis[73–75], and our study establishes a causal relationship between GSH metabolism and ferroptosis in the skeletal system. Further studies are required to understand how transferred mitochondria alter the GSH metabolism of recipient cells, and techniques need to be developed to accurately regulate the process of mitochondrial transfer.

GC is widely used to treat immune-related disorders. Our studies indicate that mitochondrial transfer occurs in bone marrow frequently under healthy conditions. However, after GC treatment, the frequency of mitochondrial transfer from osteolineage cells to myeloid lineage and lymphoid lineage cells markedly decreased. Transferred mitochondria could regulate the immune response by enhancing macrophage phagocytosis for antimicrobial effects in acute respiratory distress syndrome (ARDS)[76] or promoting inflammation[38,77,78], indicating the close relation between impaired mitochondrial transfer and the immunosuppressive effects induced by GC. Furthermore, we demonstrated that GC impairs mitochondrial transfer from osteolineage cells to myeloid cells. Targeting the GSH metabolism altered by osteolineage cell-derived mitochondria using BSO alleviated the progression of GIOP. These observations implicate a reduction in mitochondrial transfer as a feature of GC-induced metabolic disorders in the skeletal system.

Besides, to be consistent with the metabolic changes, diurnal exposure pattern and kinetics of clinical glucocorticoid therapy, we established GIOP mice model using GC drinking water treatment according to the study from Gasparini, Sylvia J et al.[53]. GC in drinking water may lead to the high variation as the difficulty in controlling the volume of water ingested and thus the dose administered. However, this did not pose a significant problem in our study as we recorded and compared the water intake of different groups, which showed no significant differences. Furthermore, while some studies may report that female are less vulnerable to GC treatment, there were also studies reporting the GIOP model in female mice[79,80]. In our study, we observed the phenomenon of mitochondrial transfer in both male and female mice under GC treatment, and the differences in female group are slightly less than male group, which is consistent with previous studies.

In summary, our findings depict the mitochondrial transfer network in the skeletal system and reveal a paradigm to regulate coupled bone remodeling via MIRO1-mediated mitochondrial transfer from osteolineage cells to myeloid cells, thus regulating skeletal metabolic homeostasis, which also provides insights into GIOP progression. Therapeutically targeting mitochondrial transfer may represent a way to ameliorate skeletal metabolic diseases.

## Methods

### Study approval

Our research complies with all relevant ethical regulations according to protocols approved by the Animal Care and Use Committee of Shanghai Jiao Tong University affiliated to Sixth People's Hospital (No: 2021-0937). All mice were sacrificed and all animal experiments were performed under isoflurane anesthesia. And every effort was made to minimize suffering.

### Mice

*Dmp1cre* mice (marks osteocytes) were provided by J. Q. (Jerry) Feng from Texas A&M College of Dentistry, USA (Jackson Laboratory stock number, 023047). *Prrx1cre* mice (marks osteoprogenitors) (stock number, 005584), *Col1a1cre/ERT2* mice (marks osteoblastic cells) (stock number, 016241), *MitoDendra+/+* mice (stock number, 018385), *Rhot1fl/fl* mice (stock number, 031126) and *mT/mG* mice (stock number, 007676) were acquired from Jackson Laboratory. *Lyz2cre* mice were acquired from GemPharmatech (strain ID, T003822). Osteolineage mitochondria reporter mice were established by crossing *Prrx1cre*, *Col1a1cre/ERT2* and *Dmp1cre* mice with *MitoDendra+/+* mice to obtain *Prrx1cre MitoDendra+/+*, *Col1a1cre/ERT2 MitoDendra+/+* and *Dmp1cre MitoDendra+/+* mice. Myeloid lineage mitochondria reporter mice were established by crossing *Lyz2cre* mice with *MitoDendra+/+* mice to obtain *Lyz2cre MitoDendra+/+* mice. Osteolineage MIRO1 knockout mice was established by crossing *Prrx1cre*, *Col1a1cre/ERT2* and *Dmp1cre* with *Rhot1fl/fl* mice to obtain *Prrx1cre Rhot1fl/fl* mice, *Col1a1cre/ERT2 Rhot1fl/fl* mice and *Dmp1cre Rhot1fl/fl* mice. Myeloid MIRO1 knockout mice were established by crossing *Lyz2cre* mice with *Rhot1fl/fl* mice to obtain *Lyz2cre Rhot1fl/fl* mice. Osteolineage MIRO1 knockout mitochondria reporter mice were established by crossing *Prrx1cre MitoDendra+/+* with *Rhot1fl/fl* mice to obtain *Prrx1cre Rhot1fl/fl MitoDendra+/+* mice. For tamoxifen induced depletion, administer tamoxifen (approximately 75 mg/kg body weight) via intraperitoneal injection (using an ACUC approved injection procedure) once ever total of 5 consecutive days. All mice were maintained with normal chow (1010082) at the animal facility of Shanghai Jiao Tong University affiliated to Sixth People's Hospital under 12-h light/dark cycle at 20–26 °C and 40–70% humidity. All diets were prepared by Jiangsu-Xietong, Inc. (Nanjing, China). All mice were maintained on the C57BL/6J background and were used at the age of 8–16 weeks. The data for key experiments are in both sexes. Ovariectomy-induced mice model is widely used for postmenopausal osteoporosis study and is only applicable to female mice, so the experiment only used female mice.

### Flow cytometry

Bone marrow cells were obtained by flushing bone marrow of mice femurs and then filtered through 70μm nylon mesh. Erythrocyte lysis solution was used to remove red blood cells and then the isolated cells

were blocked by anti-mouse CD16/32 antibody (Biolegend, 101302) for 15 min (min). Fluorescence-conjugated antibodies were used for staining. Staining was performed at 4°C in the dark for 30 min. All the antibodies were purchased from Biolegend and listed as below: anti-CD45-Pacific Blue™ (157212), anti-CD117-PE (105808), anti-CD117-APC/Cy7 (105825), anti-CD45R-PE/Cy5 (103209), anti-CD45R-APC (103212), anti-Ly-6A/E-APC (108111), anti-Ly-6A/E-Alexa Fluor®700 (108142), anti-Ly-6C-Pacific Blue™ (128013), anti-Ly-6C-PE (128007), anti-Ly-6G-Pacific Blue™ (127611), anti-Ly-6G-PE/Cy7 (127617), anti-F4/80-Brilliant Violet 510™ (123135), anti-F4/80-Alexa Fluor®700 (123129), anti-CD11c-APC (117310), anti-lineage cocktail-Pacific Blue™ (133305), anti-CD11b-Alexa Fluor®700 (101202), anti-CD11b-APC/Cy7 (101225), anti-CD19-PE (115507), anti-CD19-PE/Cy7 (115519), anti-CD4-APC/Cy7 (100413), anti-CD4-PE/Cy5 (103209), anti-CD8-APC (100711), anti-CD8-PE/Cy7 (100721). The cells were then washed and run on the Cytometer CytoFLEX (Beckman Coulter). FlowJo software version 10.4 was used to analyze samples. 50000 events were collected for each sample.

## Fluorescence-activated cell sorting (FACS)

For BMSCs sorting, bone marrow cells were flushed and stained with anti-CD45- Pacific Blue™ (Biolegend, 157212), anti-CD11b-FITC (Biolegend, 101205), anti-Ly-6A/E-APC (Biolegend, 108111) and anti-CD29-PE (Biolegend, 102208). Stained cells were sorted on the Sony platform. CD11b⁻CD45⁻CD29⁺Sca1⁺ cells were gated as targeted cells in purifying mode. Sorted cells were cultured in mouse mesenchymal stem cell culture medium (Cyagen) and then induced for osteogenesis. Similarly, Dendra2 positive OCPs (tdTomato⁺ Dendra2⁺) and Dendra2 negative OCPs (tdTomato⁺ Dendra2⁻) were gated as targeted cells in purifying mode. Sorted cells were sent to subsequent RNA sequencing.

## Bone histomorphometry analysis

Femurs were dissected and fixed in 4% paraformaldehyde (PFA) for two days and then immersed in 10% EDTA (pH=7.2) in 4 °C for about 2 weeks for decalcification. For osteoclast analysis, decalcified femurs were embedded in paraffin and sectioned at 4 μm thickness and TRAP staining was performed to label osteoclasts. For osteoblast analysis, undecalcified femurs were embedded in plastic and sectioned at 5 μm thickness and Goldner trichrome staining was performed to label osteoblasts. For dynamic histomorphometry analysis, double calcein-labeling was used. Each mouse was injected intraperitoneally with 30 μg/gram body weight calcein (Sigma-Aldrich) on day 1 and day 7. At the day 9, mice were sacrificed, and femurs were dissected, fixed in 4% PFA for two days, embedded in plastic and sectioned at 5 μm thickness. Bioquant Osteo software (Bioquant) was used for bone histomorphometry analysis.

## Bone density measurements

Femurs and L3 lumbar were separated and fixed in 4% PFA for two days. Soft tissue of femurs and L3 lumbers was removed and immersed in 70% ethanol one day before scanned using the μCT instrument (SkyScan 1176). Relevant structure parameters of the μCT instrument were as previous reported[35]. The scanning voxel size was 9 × 9 × 9 um³. The X-ray tube potential was 50 kV and 450 uA. Integration time was 520 ms and rotation step was 0.4° for 180° scanning. CTAn micro-CT software version 1.13 (Bruker) was used for analysis. A threshold value of 75 (grayscale index) was used for all femurs and lumbar trabecular bone analysis and 110 for all cortical bone analysis. The femurs and L3 lumbar were analyzed at a resolution of 9 μm. For cortical bone analysis, the volumetric regions include 600 μm long at mid-diaphysis of the femur (300 μm extending proximally and distally from the diaphyseal midpoint between the proximal and distal growth plates). For vertebrae, the volumetric regions exclude the primary spongiosa (300 μm below the cranial and above the caudal growth plate). Morphometric data included BMD, BV/TV, Tb.N, Tb.Th, Tb.Sp, Ct.Th, and Ct.Po.

## Whole mount alcian blue/alizarin red staining

The mouse embryos (P0) were fixed and dehydrated overnight in 95% ethanol with gentle agitation after removal of skin, viscera and muscle. Then, the embryos were degreased overnight in absolute acetone and stained in 0.015% alcian blue (Sigma) /0.005% alizarin red (Sigma) in 70% ethanol overnight with gentle agitation. After staining, the embryos were washed in 70% ethanol for 30 min three times and digested using 1% KOH solution. When most of the soft tissue of the embryos was digested, 75% (vol/vol) KOH/glycerol solution was used to store the embryos and gradually replaced with glycerol. Pictures of the embryos were captured using microscope (Leica).

## Cell culture and cell-based experiments

**In vitro osteoclastogenesis assay and bone resorption assay.** Bone marrow of tibia was flushed, and the collected cells were cultured overnight by using α-MEM (Hyclone) with 10% FBS (Gibco), 100 μg/ml streptomycin (Gibco) and 100 U/ml penicillin (Gibco). Then non-adherent cells were collected and layered on Ficoll-Paque (GE Healthcare). Density gradient centrifugation was performed at 4 °C and 750g for 20 min and BMMs in the middle layer of the separation were collected and cultured for another two days in α-MEM with 10% FBS, 100 μg/ml streptomycin, 100 U/ml penicillin supplemented with 20 ng/ml M-CSF (Peprotech). For osteoclastogenesis, BMMs (2.5 × 10⁴ cells per well for 96-well plates) were induced at 37°C in a humidified incubator at 5% CO₂ by using α-MEM with 10% FBS, 100 μg/ml streptomycin, 100 U/ml penicillin, 100 ng/ml M-CSF (Peprotech) and 100 ng/ml RANKL (Peprotech). The medium was changed every 2 days. After the end of differentiation, cells were fixed and stained with Tartrate-resistant acid phosphatase (TRAP) kit according to the manufacturer's instructions (Sigma). Cells which contain more than three nuclei were regarded as mature osteoclasts and counted with ImageJ. For bone resorption assay, BMMs were seeded into a 24-well plate (1×10⁵ cells per well) with bone biomimetic synthetic surface (Corning) and induced towards osteoclastogenesis. Then, the plate was treated with NaClO solution and bone resorption area was quantified using ImageJ.

## Harvest of calvaria osteoblasts and long bone osteoblasts and osteogenic differentiation

Neonatal mice were decapitated and calvaria were separated from their skulls. Soft tissue was removed from the calvaria and then calvaria was digested using α-MEM which contains 0.1% collagenase (Roche) and 0.2% dispase (Roche) in a 37 °C constant temperature shaking table set at 10g for 10 min. Calvaria were digested for five times in total, the first four times for 10 min and the last time for 30 min, and the digestive production which contains calvaria osteoblasts from the last four times were collected and cultured in α-MEM containing 10% FBS, 100 U/ml penicillin, 100 μg/ml streptomycin. Calvaria osteoblasts were then re-plated in 96-well plates (5000 cells per well) for staining or in 6-well plates (2 × 10⁵ cells per well) for RNA isolation. After culturing the cells to 70–80% confluence, osteogenic differentiation medium (Cyagen) was replaced for further differentiation. The medium was changed every 2 days. ALP staining, alizarin red staining or RNA isolation was performed for further research.

Harvest of osteoblasts from long bone was performed as previously reported[36]. Briefly, epiphyses of long bones were cut off and bone marrow was flushed out with PBS. Then long bones were cut into small pieces of 1–2 mm² and washed with PBS and incubated with collagenase II (Sigma) with shaking to remove remaining soft tissue and adhering cells. Then, fragments were washed and cultured in complete culture medium. After confluence of the cell monolayers, fragments were removed and trypsinized and cells were replated.

## Co-culture

For coculture to investigate the mitochondrial transfer, BMMs were extracted and induced towards OCPs with M-CSF (100 ng/mL) and

RANKL (100 ng/mL) stimulation. Meanwhile, OBs were extracted and coincubated for two days and then flow cytometry was performed. For flow cytometry analysis, 12-well dish was used to coculture BMMs (1 × 10⁵/ well) and OBs (1 × 10⁵/ well). At the indicated day, cells were firstly digested with trypsin-EDTA and then detached with cell scraper. To test whether extracellular vesicles or direct contract contribute to the mitochondrial transfer, a transwell system was used. BMMs were extracted from *mT/mG* mice and induced towards OCPs with M-CSF (100 ng/mL) and RANKL (100 ng/mL) stimulation. For direct coculture, OBs extracted from *Prrx1^cre MitoDendra^+/+* mice were added and coincubated for two days. For indirect coculture, OBs extracted from *Prrx1^cre MitoDendra^+/+* mice were coincubated in the upper well with OCPs plated in lower well for two days. To examine the mitochondrial transfer during osteoclast differentiation, BMMs were extracted from *mT/mG* mice and induced towards osteoclastogenesis with M-CSF (10 ng/mL) and RANKL (30 ng/mL) stimulation. Meanwhile, at day 0, day 2 and day 4, OBs extracted from *Prrx1^cre MitoDendra^+/+* mice were coincubated with cells respectively, and then at day 1, day 3 and day 5, flow cytometry was performed.

To investigate the mitochondrial transfer from OBs to BMMs, BMMs were extracted from *mT/mG* mice and maintained with M-CSF (10 ng/mL). OBs extracted from *Prrx1^cre MitoDendra^+/+* mice were added to BMMs for one day and then super resolution fluorescent analysis was performed. To investigate the mitochondrial transfer from OBs to OCPs, BMMs were extracted from *mT/mG* mice and induced towards OCPs with M-CSF (100 ng/mL) and RANKL (100 ng/mL) stimulation for two days. And at day 2, OBs extracted from *Prrx1^cre MitoDendra^+/+* mice were added to OCPs for one day and then super resolution fluorescent analysis was performed.To investigate the mitochondrial transfer from OBs to mOCs, BMMs were extracted from *mT/mG* mice and induced towards OCPs with M-CSF (100 ng/mL) and RANKL (100 ng/mL) stimulation for five days. And at day 5, OBs extracted from *Prrx1^cre MitoDendra^+/+* mice were added to mature osteoclasts (mOCs) for one day and then super resolution fluorescent analysis was performed.

To induce osteoclastogenesis and perform bone resorption assay based on OB-OC coculture system, extracted OBs were plated in the well, and then BMMs extracted from wildtype mice were plated on the osteoblasts layer and supplemented with 1, 25-dihydroxyvitamin D3 (10 nM; Sigma) and PGE2 (1 μM; Sigma) and induced towards osteoclastogenesis. The medium was changed every 2 days. After the end of differentiation, TRAP staining and quantification were performed.

### Cell imaging and analysis

Fixed cell images were acquired on the Olympus super resolution microscope using a 40× objective with motorized correction collar. Livecell images were acquired on the Olympus super resolution microscope using a 40× objective in a temperature-controlled chamber (37 °C, 5% $CO_2$). Digital images were acquired using Olympus software. All images were assembled and analyzed using Fiji (National Institutes of Health).

### Lentivirus package

For *Rhot1* or *Mfn2* overexpression, cells were transfected with pLenti-CMV-Mfn2-BSR or pLenti-CMV-Rhot1-BSR (purchased from OBiO Technology Corp., Ltd.) with 10 μg/ml polybrene for 48 h and selected by 10 μg/mL blasticidin (Thermo Fisher, Cat# A1113902). The detailed sequences are *Rhot1* 5′-GAT ATC TCA GAG TCG GAA TTT-3′ and *Mfn2* 5′-GCT GGA CAG CTG GAT TGA TAA-3′. For Rhot1 or Mfn2 knockdown, the shRNA oligonucleotides for *Rhot1* or *Mfn2* were synthesized by Tsingke Biotechnology Co. Ltd. (Beijing, China). The detailed sequences are *Rhot1_1*: 5′-GCT CAA CTT CTT CCA GAG AAT-3′, and *Rhot1_2*: 5′-GAT ATC TCA GAG TCG GAA TTT-3′, *Mfn2_1*: 5′-GCT GGA CAG CTG GAT TGA TAA-3′, *Mfn2_2*: 5′-GGC AGT TTG AGG AGT GCA TTT-3′. After annealing at 95 degrees for 20 min, the double oligonucleotides were cloned into the lentivirus vector pLKO.1-hygro. And 293 T cells were

then cotransfected with plasmids and the lentivirus packaging plasmid (psPAX2 and pMD2.G) using Lipofectamine 3000 transfection reagents (Thermo Fisher Scientific, catalog no. L3000001) following the manufacturer's protocol. An appropriate empty vector was created for shRNA constructs. After 48 h of cotransfection, the lentiviral particles were harvested from the medium. A 0.22 μm filter was used to filter the medium. Lentivirus was harvested into Eppendorf tubes and stored at 20 °C for short-term storage or 80 °C for long-term storage.

### qPCR

Total RNA was isolated using RNeasy® Mini Kit (Qiagen). 500 ng of total RNA was reverse transcribed into cDNA using PrimeScriptTM RT Master Mix (Takara, RR036A). qPCR analyses were performed using SYBR Premix Ex TaqTM II (Takara, RR820L) and samples were run on the ABI HT7900 platform (Applied Biosystems). SYBR Green PCR conditions were 1 cycle of 95 °C for 30 s (s), and 40 cycles of 95 °C for 5 s and 34 °C for 60 s. Melting curve stage was added to check primers specificity. Relative gene expression levels were calculated using the threshold cycle ($2^{-\Delta\Delta CT}$) method. Relevant primers were listed as below: *Gapdh*: 5′-ACC CAG AAG ACT GTG GAT GG-3′and 5′-CAC ATT GGG GGT AGG AAC AC-3′; *Acp5*: 5′-TGG ACA TGA CCA CAA CCT GCA GTA-3′and 5′-TCG CAC AGA GGG ATC CAT GAA GTT-3′; *Ctsk*: 5′-CCA GTG GGA GCT ATG GAA GA-3′ and 5′-AAG TGG TTC ATG GCC AGT TC-3′; *Atp6v0d2*: 5′-ACA TGT CCA CTG GAA GCC CAG TAA-3′and 5′-ATG AAC GTA TGA GGC CAG TGA GCA-3′; *Dcstamp*: 5′-AAA ACC CTT GGG CTG TTC TT-3′ and 5′-AAT CAT GGA CGA CTC CTT GG-3′; *Oscar*: 5′-GTC AGG CTT GTT GAA TTA AAG-3′ and 5′-AAG GCA CAG GAA GGA AAT AGA G-3′; *Ocstamp*: 5′-TGG GCC TCC ATA TGA CCT CGA GTA G-3′and 5′-TCA AAG GCT TGT AAA TTG GAG GAG T-3′; *Nfatc1*: 5′-AGA TGG TGC TGT CTG GCC ATA ACT-3′ and 5′-TGG TTG CGG AAA GGT GGT ATC TCA-3′; *Col1a1*: 5′-ATA AGT CCC TTC CTG CCC AC-3′ and 5′-TGG GAC ATT TCA GCA TTG CC-3′; *Spp1*: 5′-ATG CCA CAG ATG AGG ACC TC-3′ and 5′-CCT GGC TCT CTT TGG AAT GC-3′; *Sp7*: 5′-TCG GGG AAG AAG AAG CCA AT-3′ and 5′-CAA TAG GAG AGA GCG AGG GG-3′; *Runx2*: 5′-GCC CAG GCG TAT TTC AGA TG-3′ and 5′-GGT AAA GGT GGC TGG GTA GT-3′; *Tnfrsf11*: 5′-CAG CCA TTT GCA CAC CTC AC-3′ and 5′-GTC TGT AGG TAC GCT TCC CG-3′; *Tnfrsf11b*: 5′-TTC CCG AGG ACC ACA ATG AA-3′ and 5′-TCT TCC TCC TCA CTG TGC AG-3′; *Gss*: 5′-CCC ATT CAC GCT TTT CCC CT-3′ and 5′-GGG CAG TAT AGT CGT CCT TTT TG-3′; *Gsr*: 5′-GAC ACC TCT TCC TTC GAC TAC C-3′ and 5′-CAC ATC CAA CAT TCA CGC AAG-3′; *Gclc*: 5′-GCA TCC TCC AGT TCC TGC AC-3′ and 5′-GGT CGG ATG GTT GGG GTT TG-3′; *Gclm*: 5′-AGG AGC TTC GGG ACT GTA TCC-3′ and 5′-GGG ACA TGG TGC ATT CCA AAA-3′; *Gls*: 5′-TTC GCC CTC GGA GAT CCT AC-3′ and 5′-CCA AGC TAG GTA ACA GAC CCT-3′; *Gpx4*: 5′-GCC TGG ATA AGT ACA GGG GTT-3′ and 5′-CAT GCA GAT CGA CTA GCT GAG-3′. All the primers were synthesized by Sangon Biotech company (Shanghai).

### Isolation of mitochondria from calvaria osteoblast

To obtain dissociative mitochondria from calvaria osteoblasts, Mitochondria Isolation Kit (Thermo Fisher) was used according to the manufacturer's instructions. Briefly, osteoblasts were collected in a 2.0 ml microcentrifuge tube and centrifuged at 850*g* for 5 min. The supernatant was removed and 800 μl Mitochondria Isolation Reagent A was added on ice. The tube was vortexed at medium speed for about 5 s and incubated on ice for 2 min. The cell suspension was then transferred to Dounce Tissue Grinder and homogenized on ice. Cell lysates were returned to a new tube and 1 mL of Mitochondria Isolation Reagent C was added. The tube was inverted several times gently to mix and centrifuged at 700*g* for 10 min at 4°C. The supernatant was transferred to a new 2.0 ml tube and then centrifuged at 12,000*g* for 15 min at 4 °C. Remove the supernatant and the pellet contained the isolated mitochondria. Finally, 500 μl Mitochondria Isolation Reagent C was added to the pellet and centrifuged at 12,000*g* for 5 min. Discard the supernatant and resuspend the pellet by using complete medium.

## Western blot analysis

To examine the effect of gene knockout and shRNA transfection, western blot analysis was applied to determinate the expression level of relevant protein. Protein samples were run by SDS/PAGE gel electrophoresis and then transferred to PVDF membrane (Millipore). Membranes were blocked using skim milk and then incubated with primary antibodies at 4 °C overnight with gentle agitation. Following primary antibodies were used: anti-MIRO1 (1:1000, ab211363, abcam); anti-VDAC (1:1000, 4661, Cell Signaling Technology); anti-MFN2 (1:1000, 9482, Cell Signaling Technology). After washing with TBST for 3 times every 10 min, blots were incubated with secondary HRP-conjugated antibodies for 60 min at room temperature and then washed by TBST for 3 times every 10 min. Pierce® 953 ECL Western Blotting Substrate 954 (Thermo Fisher) and digital image system (BioRad) were used to visualize the protein blots. Unprocessed blots are supplied in the source data file or at the end of the Supplementary information.

## Analysis of mitochondrial mass, membrane potential and ROS

To stain mitochondria and mitochondrial ROS, calvaria osteoblast were stained with MitoTracker Green (M7514, Thermo Fisher) or MitoSOX (M36008, Thermo Fisher) for 1 h. For mitochondria membrane potential analysis, calvaria osteoblast were stained with JC-1 dye (T3168, Thermo Fisher). Flow cytometry analysis was performed using Cytometer CytoFlex (Beckman Coulter) and analyzed using FlowJo software version 10.4.

**ELISA**. ELISA of mice serum was performed as kit instructions (Jianglai). Briefly, working standards and diluted samples were prepared and added to each well. Plates were sealed and incubated for 1 h at 37 °C. After washing three times, 100 µl enzyme-labeled reagents were added and plates were incubated for 1 h at 37 °C. Finally, TMB substrates were added and incubated for 15–30 min at 37 °C followed by Stop solution addition. Then plates were read at 450 nm within 5 min.

## Metabolomics

Cell samples were resuspended with prechilled 80% methanol by well vortex. Then the samples were melted on ice and whirled for 30 s. After the sonication for 6 min, they were centrifuged, and the supernatant was freeze-dried and dissolved with 10% methanol and sent for LC-MS/MS analysis.

UHPLC-MS/MS analyses were performed using a Vanquish UHPLC system (Thermo Fisher) coupled with an Orbitrap Q ExactiveTMHF-X mass spectrometer (Thermo Fisher). Samples were injected onto a Hypesil Gold column (100×2.1 mm, 1.9 µm) using a 12-min linear gradient at a flow rate of 0.2 mL/min. The eluents for the positive polarity mode were eluent A (0.1% FA in Water) and eluent B (Methanol). The eluents for the negative polarity mode were eluent A (5 mM ammonium acetate, pH 9.0) and eluent B (Methanol). Raw data files generated by UHPLC-MS/MS were processed using the Compound Discoverer 3.1 (CD3.1, Thermo Fisher) to perform peak alignment, peak picking, and quantitation for each metabolite. After that, peak intensities were normalized to the total spectral intensity. The normalized data was used to predict the molecular formula based on additive ions, molecular ion peaks and fragment ions. And then peaks were matched with the mzCloud (https://www.mzcloud.org/), mzVault and MassList database to obtain the accurate qualitative and relative quantitative results. Statistical analyses were performed using the statistical software R (R version R-3.4.3), Python (Python 2.7.6 version) and CentOS (CentOS release 6.6).

Metabolites were annotated using the KEGG database (https://www.genome.jp/kegg/pathway.html), HMDB database (https://hmdb.ca/metabolites) and LIPIDMaps database (http://www.lipidmaps.org/). Principal components analysis (PCA) and Partial least squares discriminant analysis (PLS-DA) were performed at metaX. We applied univariate analysis (*t* test) to calculate the statistical significance (P-value). The metabolites with VIP > 1 and P-value < 0.05 and fold change ≥2 or FC ≤ 0.5 were considered to be differential metabolites. Volcano plots were used to filter metabolites of interest which based on log2(FoldChange) and -log10 (*p*-value) of metabolites by ggplot2 in R language. For clustering heatmaps, the data were normalized using z-scores of the intensity areas of differential metabolites and were ploted by Pheatmap package in R language. The functions of these metabolites and metabolic pathways were studied using the KEGG database. The metabolic pathways enrichment of differential metabolites was performed, when ratio was satisfied by x/n > y/N, metabolic pathway was considered as enrichment, when P-value of metabolic pathway <0.05, metabolic pathway was considered as statistically significant enrichment. The data were deposited into the MetaboLights (identifier: MTBLS7340).

## RNA sequencing

For preparation of OCPs with coculture, BMMs were extracted from m*T/mG* mice and induced toward OCPs with M-CSF (100 ng/mL) and RANKL (100 ng/mL) stimulation. Meanwhile, OBs extracted from *Prrx1cre MitoDendra+/+* mice were added and coincubated for two days, and then, cell sorting was performed. tdTomato+Dendra2+ cells were sorted as mtD2pos cells. tdTomato+Dendra2- cells were sorted as mtD2neg cells. For preparation of OCPs without coculture (Ctrl), BMMs were extracted from *mT/mG* mice and induced toward OCPs with M-CSF (100 ng/mL) and RANKL (100 ng/mL) stimulation for two days. Sorted mtD2neg and mtD2pos OCPs after coculture and OCPs without coculture (Ctrl) were sent for subsequent RNA sequencing. mtD2pos and mtD2neg OCPs after coculture were sorted and purified. Unbiased RNA sequencing was performed among sorted mT/mG mtD2pos and mtD2neg OCPs and OCPs without coculture (Ctrl). RNA was extracted using Trizol reagent (Thermofisher), quantified using Bioanalyzer 2100 and purified with RNA 6000 Nano LabChip Kit (Agilent). After RNA purification, mRNA library was constructed, fragmented and amplified according to manufacturer's instructions. The samples were then loaded into the nanoarray and Illumina NovaseqTM 6000 platform was used for RNA-sequencing following the vendor's recommended protocol. Generated reads were filtered and mapped to the reference genome using HISAT2 (v2.0.4) and assembled using StringTie (v1.3.4d) with default parameters. Gffcompare software (v0.9.8) was used to merge all transcriptomes from all samples to reconstruct a comprehensive transcriptome. After merging, the expression levels of all transcripts were calculated by Stringtie and ballgown. DESeq2 software was performed to analyze different gene expression and GO functions were performed for enrichment analysis. The data were deposited into the GEO repository (accession no. GSE222961).

Similarly, myeloid cells which have received or not osteoprogenitor mitochondria were sorted and performed unbiased RNA sequencing to investigate the impact of mitochondrial transfer from osteoprogenitor cells to myeloid cells in vivo. The data were deposited into the GEO repository (accession no. GSE248687).

Besides, after osteolineage cells derived mitochondria were transplanted into different stage osteoclastic lineage cells (BMMs, OCPs and mOCs), RNA was extracted and subsequent RNA-sequencing was performed as previously described. The data were deposited into the GEO repository (accession no. GSE225414).

## GSH quantification

BMMs were seeded in a 6-well plate. Cells were given indicated treatment and lysed. The supernatant was collected and the total GSH and GSSG levels were determined by GSH and GSSG Assay Kit (DOJINDO, G263) following the manufacturer's instructions.

## Cell viability test

The cell counting kit-8 (CCK-8) assay kit (DOJINDO, CK04) was used strictly according to the instructions to evaluate cell viability. Briefly, BMMs were treated as indicated for 24 h, and the culture medium was then replaced with CCK-8 working solution containing 10% CCK-8 reagent. The cells were cultured at 37 °C for 1 h. The absorbance at the wavelength of 450 nm of each well was measured by a microplate reader (Bio-Rad).

## Ferroptosis relevant staining

BMMs isolated from wild-type mice were seeded in 24-well plates with $1 \times 10^5$ cells/well and given indicated treatment. Liperfluo staining, TUNEL assay and FerroOrange staining were performed according to the manufacturer's instructions to test ferroptosis. FerroOrange staining kit (F374) and Liperfluo staining kit (L248) were purchased from DOJINDO laboratories and TUNEL assay kit was purchased from Beyotime (C1086). The images were captured using confocal microscopy (Leica).

## Seahorse analysis

The XFe96 Extracellular Flux Analyzer (Seahorse Bioscience) was used to perform extracellular flux analysis. BMMs were seeded in XFe96 cell culture plates at $1 \times 10^5$ cells/well and induced for osteoclastogenesis. Half of wells of BMMs were treated with freshly isolated mitochondria from calvarial osteoblasts and incubated overnight. Oxygen consumption rate (OCR) measurement was performed in XF media (non-buffered DMEM) supplemented with 11 mM glucose, 2 mM L-glutamine and 1 mM sodium pyruvate under basal conditions and in response to mitochondria inhibitors: 1.5 μM oligomycin, 50 nM trifluoromethoxy carbonylcyanide phenylhydrazone (FCCP) and 1 μM rotenone + 5 μM antimycin A (Sigma). Relevant parameters including OCR, basal oxygen consumption and maximal oxygen consumption were calculated. XFe-96 software and GraphPad Prism software were used to analyze the data set.

**TEM.** In order to observe the morphology of mitochondria, TEM was used. Suspended cells were collected and centrifuged at the speed of $40g$ for 5 min and then fixed with glutaraldehyde solution (2.5%, Electron Microscope Grade) at 4 °C. Samples were dehydrated in 70%, 90% and 100% alcohol and then put in fresh Epon and embedded in Epon overnight. Tecnai G2 Spirit 120 kv microscope (FEI, Hong Kong) was used to photograph mitochondria.

## GC treatment and BSO treatment in vivo

For GC treatment, mice were given 50 μg corticosterone (CS) /mL drinking water (Sigma, C2505) for 8 weeks as previously reported[53]. For BSO treatment, mice were injected intraperitoneally with BSO (8 mmol/kg) (Sigma, 19176) at 4 days intervals for 8 weeks[57]. After 4 weeks, mice were sacrificed for further analysis.

## Ovariectomy mice model and bone injury model

Mice were anesthetized with isoflurane. For ovariectomy, mice were shaved the hair, and sterilized. A midline dorsal skin incision was used, and the ovaries were excised with scissors. For sham-treated mice, the same procedure was conducted without removing the ovaries. After 8 weeks, mice were sacrificed for further analysis.

For bone injury model, femoral diaphysis was exposed and perforations 0.8 mm in diameter were then locally generated through the posterior and anterior cortices using a 21-gauge needle. According to previous research regarding cortical defect healing in normal mice, there was hematoma formation at day 3, there was new bone formation at day 7 and day 10, and the defect region underwent remodeling at day 14[81]. Thus, after 2 weeks, mice were sacrificed for further analysis.

## Statistics and reproducibility

All data were analyzed using GraphPad Prism (v8.2.1) software for statistical significance and presented as mean ± s.d. A Shapiro-Wilk test was used to test normality. The assumptions of homogeneity of error variances were tested using the $F$-test ($P > 0.05$). Unpaired two-tailed Student's $t$-test was used to determine the significance between two groups of normally distributed data. Welch's correction was used for groups with unequal variances. Mann-Whitney U test was used to determine the significance between two groups without a normal distribution. For comparisons between multiple groups with normally distributed data, ordinary one-way ANOVA test with Tukey's multiple comparisons was used. Brown-Forsythe and Welch ANOVA tests with Dunnett's T3 multiple comparisons was used for groups with unequal variances. For comparisons between multiple groups without normally distributed data, non-parametric one-way ANOVA test with Dunn's multiple comparisons was used. All inclusion/exclusion criteria were predetermined and no samples or animals were ruled out of the analysis. The experiments were randomized, and during experiments and outcome evaluation, the investigators were blinded to assignment.

## Data availability

Single cell RNA sequencing data have been deposited into GEO repository under accession code GSE202516[4]. RNA sequencing data have been deposited into GEO repository under accession codes GSE22296199 [https://www.ncbi.nlm.nih.gov/geo/query/acc.cgi?acc=GSE222961], GSE248687 and GSE225414 respectively. Metabolomics data have been deposited into MetaboLights under accession codes MTBLS7340. Data generated in this study are provided in Figshare (https://figshare.com/s/e31ddd1b657b264581d1). Source data are provided with this paper.

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

## Acknowledgements

The authors thank Professor Yueping Shen for his kindness in the guidance of statistical analysis. We acknowledge Mr. Zihao Li and Ms. Danhong Qiu for their assistance with the schematic diagrams. This work was supported by National Natural Science Foundation of China (82002339 to J.J.G., 82072417 to Y.S.G., 81820108020 to C.Q.Z.), Shanghai Frontiers Science Center of Degeneration and Regeneration in Skeletal System (BJ1-9000-22-4002), Shanghai Municipal Health Commission Key Priority Discipline Project, Shanghai Spinal Disease and Trauma Orthopedics Research Center (2022ZZ01014), and Shanghai Municipal Hospital Orthopedic Specialist Alliance (SHDC22021308).

## Author contributions

J.J.G., Y.S.G. and C.Q.Z. conceived, designed, and supervised the study. P.D., C.A.G., J.Z., J.L.M. performed the experiment and analyzed the data. G.L., D.L.L., H.L., P.L., M.Y., B.Q.W., Y.F.L., X.Y.P., C.Y.J., J.M.Y., Y.G.H. and M.H.Z. provided suggestions. P.D., C.A.G. J.L.L., J.Z. wrote the manuscript, J.J.G., Y.S.G., M.H.Z. and C.Q.Z. revised the manuscript.

## Competing interests

The authors declare no competing interests.
