## [Peer Review File · Nature Communications]

Mitochondria from osteolineage cells regulate myeloid cells mediated bone resorptionEditorial Note:

This manuscript has been previously reviewed at another journal that is not operating a transparent peer review scheme. This document only contains reviewer comments and rebuttal letters for versions considered at *Nature Communications*.

REVIEWER COMMENTS

Reviewer #1 (Remarks to the Author):

Although authors addressed most my concerns, there are still important information missing:

- 1) Coculture: the main approach to test mitochondria transfer in this manuscript is in vitro coculture system, but the detailed information is still missing. For example: macrophage and osteoblast No.? which kind culture dish? How were cells lifted for FACS analysis? More importantly, what is the gating strategy for FACS analysis of coculture experiment? Tdtomato-/Dendra2+ cell population (Osteoblast) should be shown too.
- 2) How were the Col1 CreERT2 mice were treated before analysis (Supple fig 2)? With Tamoxifen? Dose and time should be provided.

Reviewer #2 (Remarks to the Author):

The authors have responded appropriately to my main comments following their submission to Nature Metabolism and addressed the main issues with relevant experiments. The manuscript is greatly improved by comparison to the original version submitted to Nature Metabolism. In my opinion this new version deserves publication in Nature Communications.

I have minor comments. Western blot expression of Miro-1 should be quantified (Figures 12a, 14a and b, 15 a).

The method for inducing Miro1 expression in OB (results shown in figure 4f) should be detailed. There is no information on the lentiviral vector used (sequence, manufacturer, control vector) and the particle concentration used.

I'm not convinced that respiration of transferred mitochondria isn't a role in osteoclastic differentiation given the distribution of experimental points shown in Figure18b. Although this is not the main point of the work, it would be helpful to increase the number of experiments in order to have a clear answer to this question.

Reviewer #3 (Remarks to the Author):

The response to my initial comments about the manuscript have been addressed. However, there are still concerns.

1. A basic understanding of GIOP appears to still be missing.

As stated, the experimental model used is not ever been consistent leaving concerns related to the number of animals needed.

2. Mitochondrial transfer between cells, including bone cells is not novel, and this reviewer wonders how sex and age may affect it.

Some speculation would be useful.

3. The statistics are complete now, but rather confusing, why not just use non-parametric tests with the appropriate post-hocs.

4. Additional comments regarding sex differences might be useful related to all of the experiments.

Point-by-point response to reviewer's comments

We gratefully thank the editor and all reviewers for their time spent making constructive remarks and useful suggestions to help us improve the manuscript. Each suggested revision and comment were accurately incorporated and considered. Below the comments of the reviewers are response point by point and the revisions are indicated.

Reviewer #1:

General comment:

Although authors addressed most my concerns, there are still important information missing:

Response to General Comment:

We are grateful for the Reviewer's positive comments. Additional analysis was conducted to address reviewers' comments and highlighted in our revised manuscript.

Comments:

1. Coculture: the main approach to test mitochondria transfer in this manuscript is in vitro coculture system, 1) but the detailed information is still missing. For example: macrophage and osteoblast No.? which kind culture dish? How were cells lifted for FACS analysis? 2) More importantly, what is the gating strategy for FACS analysis of coculture experiment? 3) Tdtomato-/Dendra2+ cell population (Osteoblast) should be shown too.

Response to Comment 1:

We thank the Reviewer's suggestions.

1) We have added the detailed information in our revised manuscript (**Line 849-851**) and listed as following:

For flow cytometry analysis, 12-well dish was used to coculture BMMs (1×10^5 / well) and OBs (1×10^5 / well). At the indicated day, cells were firstly digested with trypsin-EDTA and then detached with cell scraper.

2) We have provided gating strategy in our revised manuscript (**Supplementary Fig. 3a**) and as following (**Response Fig. 1a**).

3) We have shown Tdtomato⁻/Dendra2⁺ cell population (Osteoblast) in co-cultured panel in **Response Fig. 1a**. We firstly distinguish the OBs and OCPs according to the tdTomato signal, then we distinguished the mitochondria transferred OCPs according to the Dendra2 signal.

2. How were the *Col1 CreERT2* mice were treated before analysis (Supple fig 2)? With Tamoxifen? Dose and time should be provided.

Response to Comment 2:

The *Col1 CreERT2* mice were treated with tamoxifen before analysis. For tamoxifen induced depletion, administer tamoxifen (approximately 75mg/kg body weight) via intraperitoneal injection (using an ACUC approved injection procedure) once ever total of 5 consecutive days. This information has been updated in the revised manuscript (**Line 724-727**).

Reviewer #2:

General comment:

The authors have responded appropriately to my main comments following their submission to Nature Metabolism and addressed the main issues with relevant experiments. The manuscript is greatly improved by comparison to the original version submitted to Nature Metabolism. In my opinion this new version deserves publication in Nature Communications.

Response to General Comment:

We thank the Reviewer's positive comments. We have performed additional analysis to answer reviewer's comments and highlighted in our revised manuscript.

Comments:

1. I have minor comments. Western blot expression of Miro-1 should be quantified (Figures 12a, 14a and b, 15 a).

Response to Comment 1:

We have quantified the expression of Miro1 in Figures 12a, 14a and b, 15a, which are listed as following (**Response Fig. 2a, Response Fig. 3a-b, Response Fig. 4a**):

Response Fig. 2. a, Western blot and quantification of Miro1 expression in OBs with or without direct coculture and indirect coculture with OCPs under RANKL stimulation. BMMs were extracted and induced toward OCPs with M-CSF (100 ng/mL) and RANKL (100 ng/mL) stimulation with or without direct coculture of OBs and indirect coculture of OBs for two days. Data are presented as the mean \pm s.d. *P* values were determined by ordinary one-way ANOVA with Tukey's multiple comparisons test.

Response Fig. 3. a, Western blot and quantification demonstrating the knockout of Miro1 in osteoprogenitors in vivo ($n = 3$ per group). **b**, Western blot and quantification demonstrating the knockout of Miro1 in osteoblast in vitro in *Prrx1^{Cre} Rhot1^{fl/fl}* mice ($n = 3$ per group). Data are presented as the mean \pm s.d., with biologically individual data points shown. *P* values were determined by unpaired two-tailed Student's *t*-test.

Response Fig. 4. a, Western blot and quantification of sorted osteoprogenitors after GC treatment in vivo demonstrating significant downregulation of Miro1 ($n = 3$ per group). Data are presented as the mean \pm s.d., with biologically individual data points shown. *P* values were determined by unpaired two-tailed Student's *t*-test with Welch's correction.

2. The method for inducing Miro1 expression in OB (results shown in figure 4f) should be detailed. There is no information on the lentiviral vector used (sequence, manufacturer, control vector) and the particle concentration used.

Response to Comment 2:

We thank the Reviewer for pointing out this issue. We now have updated this information in our revised manuscript (**Line 893-910**) and as following:

For *Rhot1* or *Mfn2* overexpression, cells were transfected with pLenti-CMV-Mfn2-BSR or pLenti-CMV-Rhot1-BSR (purchased from OBiO Technology Corp., Ltd.) with 10 µg/ml polybrene for 48 h and selected by 10 µg/mL blasticidin (Thermo Fisher, Cat# A1113902). The detailed sequences are *Rhot1* 5'-GAT ATC TCA GAG TCG GAA TTT-3' and *Mfn2* 5'-GCT GGA CAG CTG GAT TGA TAA-3'. For Rhot1 or Mfn2 knockdown, the shRNA oligonucleotides for *Rhot1* or *Mfn2* were synthesized by Tsingke Biotechnology Co. Ltd. (Beijing, China). The detailed sequences are *Rhot1_1*: 5'-GCT CAA CTT CTT CCA GAG AAT-3', and *Rhot1_2*: 5'-GAT ATC TCA GAG TCG GAA TTT-3', *Mfn2_1*: 5'-GCT GGA CAG CTG GAT TGA TAA-3', *Mfn2_2*: 5'-GGC AGT TTG AGG AGT GCA TTT-3'. After annealing at 95 degrees for 20 minutes, the double oligonucleotides were cloned into the lentivirus vector pLKO.1-hygro. And 293T cells were then cotransfected with plasmids and the lentivirus packaging plasmid (psPAX2 and pMD2.G) using Lipofectamine 3000 transfection reagents (Thermo Fisher Scientific, catalog no. L3000001) following the manufacturer's protocol. An appropriate empty vector was created for shRNA constructs. After 48 hours of cotransfection, the lentiviral particles were harvested from the medium. A 0.22 µm filter was used to filter the medium. Lentivirus was harvested into Eppendorf tubes and stored at 20°C for short-term storage or 80°C for long-term storage.

3. I'm not convinced that respiration of transferred mitochondria isn't a role in osteoclastic differentiation given the distribution of experimental points shown in Figure 18b. Although this is not the main point of the work, it would be helpful to increase the number of experiments in order to have a clear answer to this question.

Response to Comment 3:

We have increased the number of experiments and demonstrated that when transplanted with respiration impaired mitochondria from osteoblasts, the inhibitory effect of osteoclast differentiation was aborted (**Response Fig. 5a, b**). These results demonstrated that the respiration of transferred osteoblast mitochondria is important to regulate osteoclastic differentiation.

Response Fig. 5. a, b, TRAP staining of in vitro osteoclastogenesis after transplanted with mitochondria or A&R pretreated mitochondria (**a**) and quantitative analysis (**b**) of TRAP-positive cells (nucleus > 3) per well ($n = 8$ per group). Scale bar, 250 µm. Data are presented as the mean \pm s.d., with biologically individual data points shown. P values were determined by ordinary one-way ANOVA with Tukey's multiple comparisons test (**b**).

Reviewer #3:

General comment:

The response to my initial comments about the manuscript have been addressed. However, there are still concerns.

Response to General Comment:

We appreciate the Reviewer's positive comments. Based on the reviewer's comments, we have performed additional analysis and highlighted the important parts in our revised manuscript.

Comments:

1. A basic understanding of GIOP appears to still be missing.

As stated, the experimental model used is not ever been consistent leaving concerns related to the number of animals needed.

Response to Comment 1:

We have added our understanding of GIOP in our revised manuscript and listed as following:

Line 505-508: Glucocorticoid (GC) medications, such as prednisone or cortisone, are commonly prescribed for various inflammatory and autoimmune conditions¹, and the main cause of GIOP. Besides primary osteoporosis, GIOP represents the most common type of secondary osteoporosis^{2,3}, which is also closely linked with energy metabolism⁴.

Line 539-544: In addition to suppress bone formation, GC treatment promotes bone resorption during the initial phase^{5, 6} by regulating the supply and lifespan of osteoclasts⁷. While there were studies investigating the mechanism from the aspect of bone formation and resorption, there were no efficient therapies treating GIOP. Targeting the crosstalk between osteoblasts and osteoclasts could be a promising strategy for the treatment of GIOP.

To be consistent with the metabolic changes, diurnal exposure pattern and kinetics of clinical glucocorticoid therapy, we established GIOP mice model using GC drinking water treatment according to the study from Gasparini, Sylvia J et al.⁸ (Steroids 2016, 116, 76-82. PMID: 27815034). And the method is safe, inexpensive, easily adjustable, non-invasive and avoids operative stress to the animals. In their study, eight-week-old mice received 25-100 µg corticosterone (CS) /mL for 4 weeks via the drinking water, and we now have performed GIOP mice model using two time points (4 weeks and 8 weeks) and increased our animal number from $n=4-6$ per group to $n=6-7$ per group. Both experiments demonstrated obvious bone loss after GC drinking water treatment (**Response Fig. 6a-e, Response Fig. 7a-e**) and we have revised our conclusion that mitochondria transfer from osteolineage cells to myeloid cells may at least in part contribute to the GIOP progression (**Fig. 7f-h, Supplementary Fig. 10j-k**).

Response Fig. 6. a-b, Representative μ CT reconstructed images of male mouse femurs without GC treatment (Ctrl), GC-treated male mice with vehicle treatment (GC + vehicle) and GC-treated male mice with BSO treatment (GC + BSO) for 8 weeks (**a**) and trabecular microstructural parameters (BMD, BV/TV, Tb.N, Tb.Sp and Tb.Th) (**b**) and cortical microstructural parameters (Ct.Th and Ct.Po) (**c**) derived from μ CT analysis, demonstrating that GSH depletion by BSO treatment attenuated the GC induced bone loss ($n = 6-7$ per group). **d, e**, TRAP staining of male mouse femurs at 8 weeks (**d**) and histomorphometry analysis of N.Oc/BS and Oc.S/BS (**e**), demonstrating that BSO administration inhibited osteoclast activity in the GIOP mouse model ($n = 6-7$ per group). Scale bar, 50 μ m. Data are presented as the mean \pm s.d., with biologically individual data points shown. P values were determined by nonparametric ANOVA with Dunn's multiple comparisons test (BV/TV, Tb.Sp of **b**), and ordinary one-way ANOVA test with Tukey's multiple comparisons (**b, c, e**).

2. Mitochondrial transfer between cells, including bone cells is not novel, and this reviewer wonders how sex and age may affect it.

Some speculation would be useful.

Response to Comment 2:

Sex exerts a profound impact on physiological osteogenesis as well as bone metabolic diseases. Significant differences were observed between females and males including the growth of bone, peak bone mass, bone mineralization and the prevalence of bone metabolism related diseases^{9, 10}. Various pathologies, as well as sensitivity to therapeutics, can be attributed to mitochondrial bioenergetics and oxidative stress, which may be involved in sex dimorphism¹¹. For example, previous study has reported that sex differences in heart mitochondria regulate diastolic dysfunction¹². Therefore, we inferred that sex may also play an important role in mitochondria transfer between bone cells, which may be the underlying mechanism of sex dimorphism in bone. On the other hand, a hallmark of aging is the weakening of bones, which become more

fragile as getting older. Our previous study has demonstrated impaired mitochondria transfer between osteocytes during the aging process with the decreased expression of transfer-related proteins¹³⁻¹⁵, therefore, aging may exert the important role in the mitochondria transfer.

3. The statistics are complete now, but rather confusing, why not just use non-parametric tests with the appropriate post-hocs.

Response to Comment 3:

The statistical analysis was employed according to published study¹⁶⁻¹⁸(Nat Metab 5, 821–841 (2023). PMID: 37188819. Nat Metab. 2023;5(1):129-146. PMID: 36635449. J Biol Chem. 2020;295(1):69-82. PMID: 31740582) and listed as following (***Line 1126-1136***):

A Shapiro-Wilk test was used to test normality. The assumptions of homogeneity of error variances were tested using the F-test ($P > 0.05$). Unpaired two-tailed Student's t-test was used to determine significance between two groups of normally distributed data. Welch's correction was used for groups with unequal variances. Mann-Whitney U test was used to determine significance between two groups without a normal distribution. For comparisons between multiple groups with normally distributed data, ordinary one-way ANOVA test with Tukey's multiple comparisons was used. Brown-Forsythe and Welch ANOVA tests with Dunnett's T3 multiple comparisons was used for groups with unequal variances. For comparisons between multiple groups without normally distributed data, non-parametric one-way ANOVA test with Dunn's multiple comparisons was used.

And indeed, we performed non-parametric analysis with appropriate post-hocs according to the Reviewer's suggestions (Mann-Whitney U test for two groups and non-parametric one-way ANOVA test with Dunn's multiple comparisons for multiple groups without normally distributed data).

4. Additional comments regarding sex differences might be useful related to all of the experiments.

Response to Comment 4:

Sex exerts a profound impact on physiological osteogenesis as well as bone metabolic diseases and significant differences were observed between females and males including the growth of bone, peak bone mass, bone mineralization and the prevalence of bone metabolism related diseases^{9, 10}. Various pathologies, as well as sensitivity to therapeutics, can be attributed to mitochondrial bioenergetics and oxidative stress, which may be involved in sex dimorphism¹¹. Different mitochondrial compositions and functions have been described in different tissues, including liver, adipocytes, neurons, brain, skeletal muscle, and heart^{12, 19, 20}. Oxygen and antioxidant capacities, calcium retention capacities, protein content and specific activity of mitochondria are all involved in mitochondrial sex specificity¹¹. Therefore, other mitochondrial functions like mitochondria transfer may also differ between males and females but have not been

investigated so far. Further studies are thus needed to decipher the sex specificity of mitochondria transfer.

Reference

1. Raterman, H. G.; Bultink, I. E. M.; Lems, W. F., Current Treatments and New Developments in the Management of Glucocorticoid-induced Osteoporosis. *Drugs* **2019**, *79* (10), 1065-1087.
2. Briot, K.; Roux, C., Glucocorticoid-induced osteoporosis. *RMD Open* **2015**, *1* (1), e000014.
3. Buckley, L.; Guyatt, G.; Fink, H. A.; Cannon, M.; Grossman, J.; Hansen, K. E.; Humphrey, M. B.; Lane, N. E.; Magrey, M.; Miller, M.; Morrison, L.; Rao, M.; Robinson, A. B.; Saha, S.; Wolver, S.; Bannuru, R. R.; Vaysbrot, E.; Osani, M.; Turgunbaev, M.; Miller, A. S.; McAlindon, T., 2017 American College of Rheumatology Guideline for the Prevention and Treatment of Glucocorticoid-Induced Osteoporosis. *Arthritis Rheumatol* **2017**, *69* (8), 1521-1537.
4. Magomedova, L.; Cummins, C. L., Glucocorticoids and Metabolic Control. *Handb Exp Pharmacol* **2016**, *233*, 73-93.
5. Chotiyarnwong, P.; McCloskey, E. V., Pathogenesis of glucocorticoid-induced osteoporosis and options for treatment. *Nat Rev Endocrinol* **2020**, *16* (8), 437-447.
6. Teitelbaum, S. L., Glucocorticoids and the osteoclast. *Clin Exp Rheumatol* **2015**, *33* (4 Suppl 92), S37-9.
7. Manolagas, S. C., Steroids and osteoporosis: the quest for mechanisms. *J Clin Invest* **2013**, *123* (5), 1919-21.
8. Gasparini, S. J.; Weber, M. C.; Henneicke, H.; Kim, S.; Zhou, H.; Seibel, M. J., Continuous corticosterone delivery via the drinking water or pellet implantation: A comparative study in mice. *Steroids* **2016**, *116*, 76-82.
9. Baxter-Jones, A. D. G.; Jackowski, S. A., Sex differences in bone mineral content and bone geometry accrual: a review of the Paediatric Bone Mineral Accrual Study (1991-2017). *Ann Hum Biol* **2021**, *48* (6), 503-516.
10. da Silva, J. A.; Porto, A., [Sex hormones and osteoporosis: a physiological perspective for prevention and therapy]. *Acta Med Port* **1997**, *10* (10), 689-95.
11. Ventura-Clapier, R.; Moulin, M.; Piquereau, J.; Lemaire, C.; Mericskay, M.; Veksler, V.; Garnier, A., Mitochondria: a central target for sex differences in pathologies. *Clin Sci (Lond)* **2017**, *131* (9), 803-822.
12. Cao, Y.; Vergnes, L.; Wang, Y. C.; Pan, C.; Chella Krishnan, K.; Moore, T. M.; Rosa-Garrido, M.; Kimball, T. H.; Zhou, Z.; Charugundla, S.; Rau, C. D.; Seldin, M. M.; Wang, J.; Wang, Y.; Vondriska, T. M.; Reue, K.; Lusis, A. J., Sex differences in heart mitochondria regulate diastolic dysfunction. *Nat Commun* **2022**, *13* (1), 3850.
13. Gao, J.; Qin, A.; Liu, D.; Ruan, R.; Wang, Q.; Yuan, J.; Cheng, T. S.; Filipovska, A.; Papadimitriou, J. M.; Dai, K.; Jiang, Q.; Gao, X.; Feng, J. Q.; Takayanagi, H.; Zhang, C.; Zheng, M. H., Endoplasmic reticulum mediates mitochondrial transfer within the osteocyte dendritic network. *Sci Adv* **2019**, *5* (11),

eaaw7215.

14. Liu, D.; Gao, Y.; Liu, J.; Huang, Y.; Yin, J.; Feng, Y.; Shi, L.; Meloni, B. P.; Zhang, C.; Zheng, M.; Gao, J., Intercellular mitochondrial transfer as a means of tissue revitalization. *Signal Transduct Target Ther* **2021**, *6* (1), 65.
15. Zhou, H.; Zhang, W.; Li, H.; Xu, F.; Yinwang, E.; Xue, Y.; Chen, T.; Wang, S.; Wang, Z.; Sun, H.; Wang, F.; Mou, H.; Yao, M.; Chai, X.; Zhang, J.; Diarra, M. D.; Li, B.; Zhang, C.; Gao, J.; Ye, Z., Osteocyte mitochondria inhibit tumor development via STING-dependent antitumor immunity. *Sci Adv* **2024**, *10* (3), eadi4298.
16. Lin, L.; Guo, Z.; He, E.; Long, X.; Wang, D.; Zhang, Y.; Guo, W.; Wei, Q.; He, W.; Wu, W.; Li, J.; Wo, L.; Hong, D.; Zheng, J.; He, M.; Zhao, Q., SIRT2 regulates extracellular vesicle-mediated liver-bone communication. *Nat Metab* **2023**, *5* (5), 821-841.
17. Zhang, K.; Wang, Y.; Chen, S.; Mao, J.; Jin, Y.; Ye, H.; Zhang, Y.; Liu, X.; Gong, C.; Cheng, X.; Huang, X.; Hoeft, A.; Chen, Q.; Li, X.; Fang, X., TREM2(hi) resident macrophages protect the septic heart by maintaining cardiomyocyte homeostasis. *Nat Metab* **2023**, *5* (1), 129-146.
18. Nishizawa, H.; Matsumoto, M.; Shindo, T.; Saigusa, D.; Kato, H.; Suzuki, K.; Sato, M.; Ishii, Y.; Shimokawa, H.; Igarashi, K., Ferroptosis is controlled by the coordinated transcriptional regulation of glutathione and labile iron metabolism by the transcription factor BACH1. *J Biol Chem* **2020**, *295* (1), 69-82.
19. Yin, L.; Luo, M.; Wang, R.; Ye, J.; Wang, X., Mitochondria in Sex Hormone-Induced Disorder of Energy Metabolism in Males and Females. *Front Endocrinol (Lausanne)* **2021**, *12*, 749451.
20. Chella Krishnan, K.; Vergnes, L.; Acin-Perez, R.; Stiles, L.; Shum, M.; Ma, L.; Mouisel, E.; Pan, C.; Moore, T. M.; Peterfy, M.; Romanoski, C. E.; Reue, K.; Bjorkegren, J. L. M.; Laakso, M.; Liesa, M.; Lusis, A. J., Sex-specific genetic regulation of adipose mitochondria and metabolic syndrome by Ndufv2. *Nat Metab* **2021**, *3* (11), 1552-1568.

REVIEWERS' COMMENTS

Reviewer #1 (Remarks to the Author):

The authors addressed all my concerns and it is OK for acceptance.

Reviewer #2 (Remarks to the Author):

Although the authors provided the Western blot quantification and the role of respiration of transferred mitochondria in their rebuttal letter, I requested in the previous revision, these new data have not been included in the novel revised forms of figures. The authors should add these new data to their respective figures. Apart from these minor points, I think that this is a manuscript that deserves to be published in Nature Communications.

Reviewer #3 (Remarks to the Author):

The investigators attempted to answer all of my initial concerns. However, this reviewer has spent many years studying osteoporosis induced by GCs in mice and the methods used, GCs in water is very tricky and the data do not seem to represent the variation in that method of supplying GCs that is well known in the field and reported in other studies. ,

In addition, female mice rarely have GC bone changes with the GC given, a better explanation for these results is really needed.

Lastly a one way ANOVA is probably not the best way to do statistics in which the hypothesis could go in either direction.

Point-by-point response to reviewer's comments

We gratefully thank the editor and all reviewers for their time spent making constructive remarks and useful suggestions to help us improve the manuscript. Each suggested revision and comment were accurately incorporated and considered. Below the comments of the reviewers are response point by point and the revisions are indicated.

REVIEWERS' COMMENTS

Reviewer #1 (Remarks to the Author):

General comment:

The authors addressed all my concerns and it is OK for acceptance.

Response to General Comment:

We are grateful for your positive comments.

Reviewer #2 (Remarks to the Author):

Comments:

Although the authors provided the Western blot quantification and the role of respiration of transferred mitochondria in their rebuttal letter, I requested in the previous revision, these new data have not been included in the novel revised forms of figures. The authors should add these new data to their respective figures. Apart from these minor points, I think that this is a manuscript that deserves to be published in Nature Communications.

Response to Comment:

We thank you for the positive comments.

Regarding the western blot quantification, we have added these new data to the respective figures (*Fig. 4n, Supplementary Fig. 4a, h, Supplementary Fig. 4c*).

Regarding the role of respiration of transferred mitochondria, as what you said that this is not the main point of the work, we actually found that these data are not closely related with our main work. Thus, we decide not to add these data to the figures.

Reviewer #3 (Remarks to the Author):

Comments:

1. The investigators attempted to answer all of my initial concerns. However, this reviewer has spent many years studying osteoporosis induced by GCs in mice and the methods used, GCs in water is very tricky and the data do not seem to represent the variation in that method of supplying GCs that is well known in the field and reported in other studies. In addition, female mice rarely have GC bone changes with the GC given, a better explanation for these results is really needed.

Response to Comment 1:

We thank you for pointing out the limitation of GIOP mice model by drinking water. We have added a paragraph in the Discussion section (**Line 688-699**) and is listed as following:

Besides, to be consistent with the metabolic changes, diurnal exposure pattern and kinetics of clinical GC therapy, we established GIOP mice model using GC drinking water treatment according to the study from Gasparini, Sylvia J et al¹. GC in drinking water may lead to the high variation as we cannot control the volume of water ingested and thus the dose administered. However, this did not pose a significant problem in our study as we recorded and compared the water intake of different groups, which showed no significant differences. Furthermore, while some studies may report that female are less vulnerable to GC treatment, there were also studies reporting the GIOP model in female mice^{2,3}. In our study, we observed the phenomenon of mitochondria transfer in both male and female mice under GC treatment and, and the differences in female group are slightly less than male group, which is consistent with previous studies.

2. *Lastly a one way ANOVA is probably not the best way to do statistics in which the hypothesis could go in either direction.*

Response to Comment 2:

In the former revision #1 and #2, we have made changes of statistical analysis according to your valuable suggestions. Now our statistical analysis is consistent with other published studies⁴⁻⁶(*Nat Metab* 5, 821–841 (2023). PMID: 37188819. *Nat Metab*. 2023;5(1):129-146. PMID: 36635449. *J Biol Chem*. 2020;295(1):69-82. PMID: 31740582).

Besides, we have consulted the statistician. He carefully examined our statistical analysis through the article and thought that while post hoc analysis you mentioned works, our statistical analysis including one-way ANOVA is appropriate and is also consistent with other published articles.

Reference

1. Gasparini, S. J.; Weber, M. C.; Henneicke, H.; Kim, S.; Zhou, H.; Seibel, M. J., Continuous corticosterone delivery via the drinking water or pellet implantation: A comparative study in mice. *Steroids* **2016**, *116*, 76-82.
2. Sato, A. Y.; Cregor, M.; McAndrews, K.; Li, T.; Condon, K. W.; Plotkin, L. I.; Bellido, T., Glucocorticoid-Induced Bone Fragility Is Prevented in Female Mice by Blocking Pyk2/Anoikis Signaling. *Endocrinology* **2019**, *160* (7), 1659-1673.
3. Sato, A. Y.; Cregor, M.; Delgado-Calle, J.; Condon, K. W.; Allen, M. R.; Peacock, M.; Plotkin, L. I.; Bellido, T., Protection From Glucocorticoid-Induced Osteoporosis by Anti-Catabolic Signaling in the Absence of Sost/Sclerostin. *J Bone Miner Res* **2016**, *31* (10), 1791-1802.
4. Lin, L.; Guo, Z.; He, E.; Long, X.; Wang, D.; Zhang, Y.; Guo, W.; Wei, Q.; He, W.; Wu, W.; Li, J.; Wo, L.; Hong, D.; Zheng, J.; He, M.; Zhao, Q., SIRT2 regulates extracellular vesicle-mediated liver-bone communication. *Nat Metab* **2023**, *5* (5), 821-841.

5. Zhang, K.; Wang, Y.; Chen, S.; Mao, J.; Jin, Y.; Ye, H.; Zhang, Y.; Liu, X.; Gong, C.; Cheng, X.; Huang, X.; Hoeft, A.; Chen, Q.; Li, X.; Fang, X., TREM2(hi) resident macrophages protect the septic heart by maintaining cardiomyocyte homeostasis. *Nat Metab* **2023**, 5 (1), 129-146.
6. Nishizawa, H.; Matsumoto, M.; Shindo, T.; Saigusa, D.; Kato, H.; Suzuki, K.; Sato, M.; Ishii, Y.; Shimokawa, H.; Igarashi, K., Ferroptosis is controlled by the coordinated transcriptional regulation of glutathione and labile iron metabolism by the transcription factor BACH1. *J Biol Chem* **2020**, 295 (1), 69-82.